# Auxin-mediated protein depletion for metabolic engineering in terpene-producing yeast

Zeyu Lu [1,2], Bingyin Peng[1,3✉], Birgitta E. Ebert[1,3], Geoff Dumsday[4] & Claudia E. Vickers [1,3,5✉]

In metabolic engineering, loss-of-function experiments are used to understand and optimise metabolism. A conditional gene inactivation tool is required when gene deletion is lethal or detrimental to growth. Here, we exploit auxin-inducible protein degradation as a metabolic engineering approach in yeast. We demonstrate its effectiveness using terpenoid production. First, we target an essential prenyl-pyrophosphate metabolism protein, farnesyl pyrophosphate synthase (Erg20p). Degradation successfully redirects metabolic flux toward monoterpene (C10) production. Second, depleting hexokinase-2, a key protein in glucose signalling transduction, lifts glucose repression and boosts production of sesquiterpene (C15) nerolidol to 3.5 g L$^{-1}$ in flask cultivation. Third, depleting acetyl-CoA carboxylase (Acc1p), another essential protein, delivers growth arrest without diminishing production capacity in nerolidol-producing yeast, providing a strategy to decouple growth and production. These studies demonstrate auxin-mediated protein degradation as an advanced tool for metabolic engineering. It also has potential for broader metabolic perturbation studies to better understand metabolism.

[1] Australian Institute for Bioengineering and Nanotechnology (AIBN), the University of Queensland, Brisbane, QLD, Australia. [2] School of Chemistry and Molecular Biosciences (SCMB), the University of Queensland, Brisbane, QLD, Australia. [3] CSIRO Future Science Platform in Synthetic Biology, Commonwealth Scientific and Industrial Research Organisation (CSIRO), Black Mountain, ACT, Australia. [4] CSIRO Manufacturing, Clayton, VIC, Australia. [5] ARC Centre of Excellence in Synthetic Biology, Queensland University of Technology, Brisbane, QLD, Australia. ✉email: b.peng@uq.edu.au; claudia.vickers@csiro.au

In metabolic engineering, optimisation of regulatory and metabolic networks is required to maximise production titre, yield, and rate. Gene deletion is an important strategy to perturb networks for systematic analysis and engineering[1–3]. However, genes that are essential for survival cannot be deleted; and for other genes, deletion results in significant metabolic burden or growth defects, which may lead to instability of engineered strains[4]. A conditional gene inactivation tool is desirable to enable loss-of-function perturbation studies in these scenarios.

The yeast *Saccharomyces cerevisiae* is an important model for Eukaryotic organisms and a common platform strain for the production of biofuels, biochemicals, and biopharmaceuticals. In yeast, several conditional methods to control protein levels have been developed, including selection for plasmid loss via a negatively selectable marker[5,6], inducible transcriptional repression[7–9], RNA interference[10,11], riboswitch-mediated translational control[12,13], and conditional protein degradation[14] such as temperature-sensitive mutants[15,16]. However, these methods have limitations when being applied in metabolic engineering and systems biology. Methods targeting transcription and translation are often ineffective for very stable proteins with low turnover rates; in these cases, the dilution effect by cell division determines the rate of depletion[17]. Consequently, reaching the loss-of-function steady state may be delayed[18], requiring prolonged cultivation to observe the physiological response. The engineered temperature-sensitive construct is based on a heat-inducible degron[16], which is inactive at 23 °C but causes the degradation of the target protein at 37 °C. Neither temperature is compatible with the cultivation temperature of 30 °C, commonly used for fermentation and physiological studies of *S. cerevisiae*.

In plants, the auxin-inducible protein degradation system is a powerful system allowing rapid protein depletion in response to the small hormone molecule auxin[19,20]. In the presence of auxin, the plant auxin receptor F-box protein TIR1/AFB forms a complex with the highly conserved S-phase kinase-associated protein (Skp), then recruits cullin (Cul), RING-box (Rbx), and E2 ubiquitin ligase proteins. Proteins containing an auxin-inducible degron (AID) sequence are recruited by this complex and polyubiquinated by the E2 ubiquitin ligase; these polyubiquinated proteins are then targeted for proteasomal degradation[21]. This system has been exploited in yeast by introducing a plant auxin receptor F-box protein and tagging the target protein with AID sequence[19,20,22–24]. Skp1p, Cul1p, Rbx1p, and E2 ubiquitin ligase are endogenous to yeast and the ubiquitin-based protein degradation pathway is universal in eukaryotes. E2 ubiquitin ligase catalyses polyubiquitination and proteasomal degradation of AID-tagged protein (Fig. 1a).

The original auxin-inducible protein degradation system suffers from auxin-independent basal degradation of AID-tagged proteins[19,20,23,24], making it problematic for controlling specific responses. Variants with reduced basal degradation have been developed recently by minimising the AID sequence[20,24], screening optimal auxin receptors[20], and/or expressing the auxin response transcription factor (ARF16) from rice (*Oryza sativa*)[25]. Previously, these auxin-inducible protein degradation systems were applied in studies of genetic functions of regulatory proteins. Its application in the area of metabolic engineering and systems biology has not been shown. We identified this system as potentially powerful tool in metabolic engineering to investigate physiological responses to loss-of-function perturbation at metabolic nodes, which are essential for growth fitness.

In this work, we reconstruct an auxin-inducible protein degradation system with reduced non-auxin-dependent degradation. We show its applicability as an engineering tool by implementing three types of metabolic engineering strategies in terpene-producing yeast: (i) flux redirection at a metabolic branch point to improve production of a monoterpene, (ii) modulation of the glucose metabolism to boost the production of the sesquiterpene nerolidol, and (iii) a switch to growth-arrest state without compromising terpene productivities.

## Results

**Reconstructing and optimising auxin-inducible protein degradation machinery.** Auxin-inducible degradation has previously been reconstructed in *S. cerevisiae* by expressing *O. sativa* TIR1 and applying a minimal AID tag (*Arabidopsis thaliana* IAA17 truncated to amino acids 71–114; IAA17[71–114], also known as AID*) to target proteins[24]. We re-validated this system by integrating a yeast enhanced green fluorescent protein (yEGFP) with an N-terminal AID* fusion under the control of a strong *TEF1* promoter, and a codon-optimised *O. sativa* TIR1 under the control of the medium-strength *ACS2* promoter[26]. The *ACS2* promoter was chosen as previous work has shown that strong constitutive expression of *TIR1* results in protein degradation in the absence of auxin[27].

To test the system, engineered yeast strains were grown to early log phase ($OD_{600} \approx 1$) and 1 mM of 1-naphthaleneacetic acid (NAA), a synthetic auxin analogue, was added ($t = 0$). The control strain, expressing yEGFP and TIR1, exhibited a slight decrease (16%) in fluorescence immediately after NAA addition, before stabilising with only a very small decrease over the length of the experiment (6 h; Fig. 1b), most likely as a result of *TEF1* promoter reduction as glucose depletes[26].

The *AID*-yEGFP* strain showed a 24% lower yEGFP fluorescence relative to the control prior to NAA addition (at 0 min; Fig. 1b). This might suggest that non-auxin-dependent degradation still occurs in TIR1/AID* system; it may also result from disruption of GFP fluorescence in the presence of the AID tag. The yEGFP construct has a slightly different nucleic acid sequence immediately after the start codon due to the presence of a restriction enzyme site that was used to clone the N-terminal tags in other constructs; this may affect the transcriptional activity of this construct relative to the others. Addition of NAA resulted in a sustained depletion until <1% of the original fluorescence could be detected by 330 min (Fig. 1b). This demonstrated effective functionality of the auxin-inducible degradation system.

To address potential causes of decreased GFP fluorescence in the AID*-tagged yEGFP strain, we investigated the effect of adding tags to the N-terminus of the AID* peptide. This approach might contribute to minimising auxin-independent degradation by insulation AID* from the terminus, which was observed previously when 9 myc epitopes was used[24]. However, repeated myc sequences could be difficult for genetic manipulation. We therefore tested other fusion partners: thioredoxin (*TRX1*), superoxide dismutase (*SOD1*), monomeric red fluorescent protein (mRFP), and metallothionein (*CUP1*).

All N-terminal fusions influenced initial fluorescence levels (Fig. 1b). A very low-level fluorescence (1.3 fold of autofluorescence) was observed in the *SOD1-AID*-yEGFP* strain; we suspect that this is due to instability of the apo-form of Sod1p, which requires a specific copper chaperone (Ccs1p) or supplementing with high copper[28]. Of the other fusion partners, Cup1p performed best, delivering significantly increased baseline fluorescence relative to the untagged control, and the rest were not helpful. After NAA addition, fluorescence decreased sharply to below 1% initial fluorescence by the end of the experiment. In order to separate the effects of the various modifications, we generated a *CUP1-AID*-yEGFP* control strain without the *TIR1* expression cassette to verify the *yEGFP* basal expressional level for the Cup1p tag. This strain had similar initial fluorescence

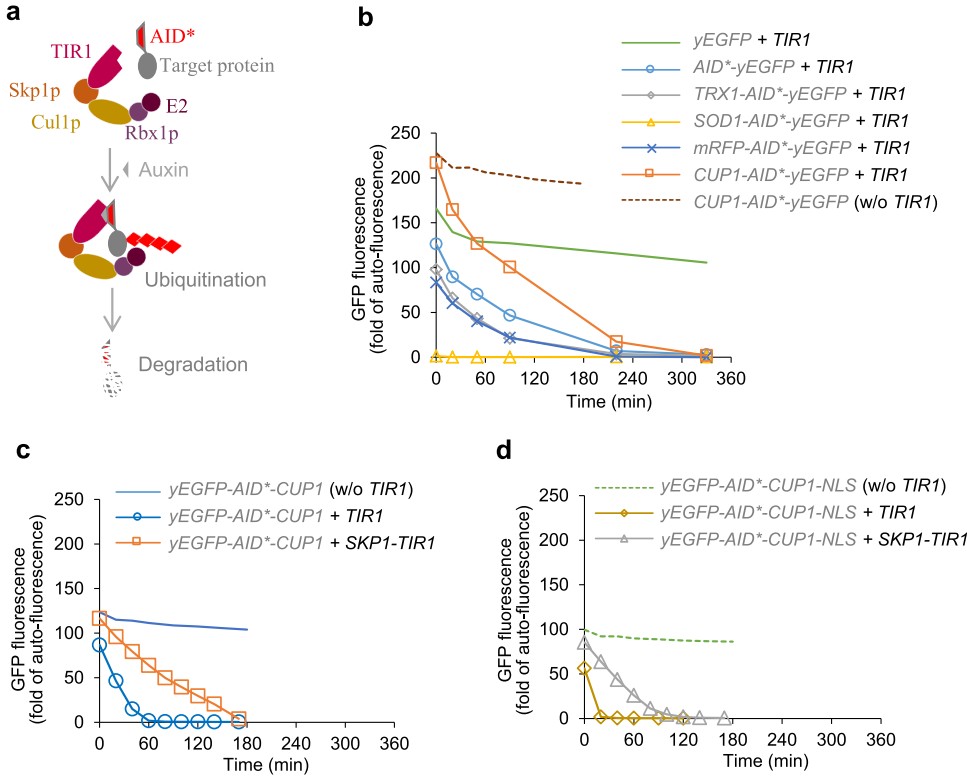

**Fig. 1 Characterising and optimising the auxin-inducible protein degradation system in yeast. a** Schematic of auxin-inducible protein degradation. **b–d** Depletion of cytosolic or nucleus-localising green fluorescent protein (GFP) with/without auxin-inducible degron tags fused in strains with/without *TIR1* or *SKP1-TIR1* expressed. N-terminal fusions (**b**), C-terminal fusions of AID* and Cup1 (**c**), and C-terminal fusions of AID*, Cup1 and the nuclear localization signal NLS (**d**). Engineered yeast strains are listed in Supplementary Table 1. The yeast cells were grown in 2-(N-morpholino)ethanesulfonic acid (MES)-buffered synthetic mineral salt medium (prepared from yeast nitrogen base, YNB) with 20 g L$^{-1}$ glucose as the carbon source to early exponential phase. GFP fluorescence was measured at $t = 0$, and synthetic auxin 1-Naphthaleneacetic acid (NAA) was immediately added to a final concentration of 1 mM to start the assay. GFP fluorescence is expressed as fold-change of exponential-phase auto-fluorescence of the reference strain. Mean values are shown ($N = 2$ or 3 independent biological replicates). Source data in **b–d** are provided as a Source Data file.

levels (prior to NAA addition), with a very slight elevation (~5%) compared to the strain expressing *TIR1*. After NAA addition, it performed very similarly to the *yEFPG + TIR1* control (Fig. 1b), demonstrating that the observed decrease in fluorescence seen in the *CUP1-AID*-yEGFP + TIR1* strain is due to auxin-inducible degradation.

A C-terminal fusion of AID*-Cup1p to yEGFP was also tested (Fig. 1c). These constructs include the restriction enzyme site found after the initial codon in original yEGFP construct (noted above). Addition of *TIR1* increased basal GFP degradation by ~30%, but resulted in extremely rapid depletion of GFP fluorescence; fluorescence decreased by >99% by 120 min.

To improve degradation efficiency, Skp1p was fused to TIR1, an approach suggested previously[29]. In contrast to the previous report, this fusion decreased degradation efficiency: it both restored basal fluorescence to control levels and slowed the degradation rate (Fig. 1c). The depletion of ~97% GFP fluorescence was achieved by 180 min.

Many yeast proteins are localised in the nucleus or simultaneously in the nucleus and cytosol. In order to test effectiveness of the engineered system for nuclear-localised proteins, we investigated localisation of the construct using a nuclear localisation sequence (NLS). The *yEGFP-AID*-CUP1* fusion was selected for further testing because it maintained basal GFP levels and provided a relatively rapid degradation phenotype. An NLS from simian virus 40 which is commonly used for nuclear localising in yeast[30,31] was fused to the C-terminal of the *yEGFP-AID*-CUP1*

construct. The control strain, *yEGFP-AID*-CUP1-NLS* without *TIR1* overexpression, behaved similarly to the other control constructs, exhibiting a small decrease in fluorescence immediately after NAA addition then maintaining relative stability over the course of the experiment (Fig. 1d). While overall fluorescence levels in all strains were slightly lower in the nuclear localisation experiments than in the cytosol experiments, basal fluorescence levels (prior to NAA addition) in both *TIR1* and *SKP1-TIR1* co-expression strains (Fig. 1d) behaved very similarly to the cytosol expression strains (Fig. 1c). The depletion of the fluorescence signal was faster in strains with *TIR1* and *SKP1-TIR1* co-expressed (Fig. 1d) compared to the cytosolic reference (Fig. 1c), achieving full fluorescence depletion at 20 and 120 min, respectively. These showed that auxin-inducible protein degradation machineries functioned for degradation of target proteins localised in both cytosol and nucleus in yeast.

For minimal potential effects on metabolism, caused by auxin-independent degradation of target enzymes, we then used the compromised *SKP1-TIR1* system, with Cup1p-AID* or AID*-Cup1p as the tag, rather than *TIR1* system that led to faster depletion of target protein, in the following studies.

**Flux redirection: degradation of farnesyl pyrophosphate improves monoterpene (limonene) production.** To demonstrate the utility of the auxin-inducible protein degradation tool as a metabolic engineering tool, we applied it to our terpene-overproducing yeast platforms[32–34]. One limitation for the

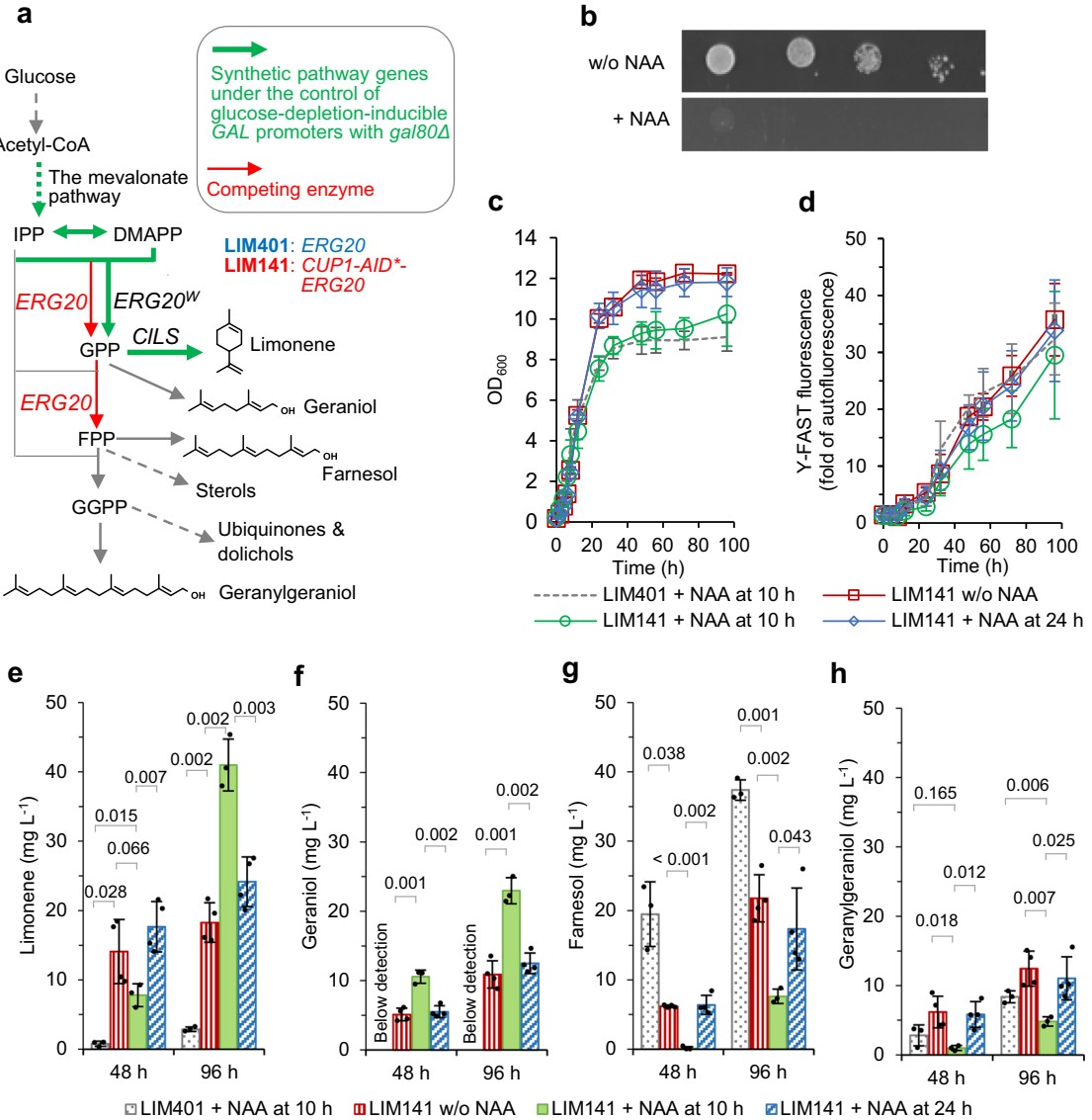

**Fig. 2 Redirecting metabolic flux towards monoterpene production via auxin-inducible degradation of farnesyl pyrophosphate synthase Erg20p.**
**a** Schematic of the engineered terpene-synthetic pathways for limonene production in strain LIM401 (*ERG20*) and LIM141 (*CUP1-AID*\*-ERG20*). Both strains have the metabolic pathway genes for limonene production overexpressed under the control of *GAL* promoters, *GAL80* disrupted, and *SKP1-TIR1* expressed for degradation of AID\*-tagged protein upon 1-Naphthaleneacetic acid (NAA) addition. IPP isopentenyl pyrophosphate, DMAPP dimethylallyl pyrophosphate, GPP geranyl pyrophosphate, FPP farnesyl pyrophosphate, GGPP geranylgeranyl pyrophosphate, *ERG20* GPP/FPP synthase, *ERG20*$^W$ ERG20 F127W mutant, GPP synthase. *CILS Citrus limon* limonene synthase, Y-FAST Yellow Fluorescence-Activating and absorption-Shifting Tag. **b** Growth of the background strain o141 (*CUP1-AID*\*-ERG20, GAL80*) on YNB-glucose agar with or without the addition of 1 mM NAA. The agar plate was incubated at 30 °C for 48 h. **c**–**h** Characterisation of limonene-producing strains LIM401 and LIM141 in two-phase flask cultivation with 20 g L$^{-1}$ glucose and dodecane overlay. Y-FAST was under the control of the *GAL10* promoter, and its fluorescence was measured after 4-hydroxy-3-methylbenzylidene rhodanine (HMBR) with final concentration 20 μM was added in yeast samples before flow cytometry assay. Y-FAST fluorescence is expressed as fold-change of exponential-phase auto-fluorescence of the reference strain GH4 (ref. [26]). Mean values ± standard deviations are shown (N = 3 or 4 independent biological replicates). Two-tailed Welch's *t*-test was used for comparing two groups, and *p* values are shown above the bars. Source data in **c**–**h** are provided as a Source Data file.

efficient production of monoterpenes (C10) in yeast is the bifunctional activity of the farnesyl pyrophosphate synthase (FPPS), Erg20p. FPPS catalyses sequential condensation of two units of isopentenyl diphosphate (IPP; C5) with one unit of dimethylallyl pyrophosphate (DMAPP, C5), forming first geranyl pyrophosphate (GPP, C10) and then farnesyl pyrophosphate (FPP, C15)[35] (Fig. 2a). GPP is not usually released from the active site before addition of the second IPP unit; consequently, there is very little free GPP available for monoterpene production[36]. To improve monoterpene production, we previously augmented the metabolic flux in a strain heterologously expressing a limonene

(C10) synthase by applying an 'N-end-rule' protein degradation method (N-degron) to destabilise Erg20p[33] (Fig. 2a). This constitutive destabilization of Erg20p significantly improved monoterpene (limonene) production, delivering 76 mg/L. However, growth in the exponential phase was very slow with a rate of ~0.15 h$^{-1}$, indicating that this modification caused significant growth defects[33]. Conditional degradation of Erg20p may allow us to separate growth and production and optimise both phases individually.

We used a mevalonate pathway augmented strain[33] to increase flux to the precursors IPP and DMAPP. In these strains, diauxie-

inducible *GAL* promoters in a *gal80Δ* background are employed to control the terpene anabolic pathway genes. This strategy improves the overall production of heterologous terpenes by avoiding the metabolic burden caused by high-level expression of these anabolic genes during the exponential growth phase[32,33,37,38]. The *SKP1-TIR1* fusion was introduced to deliver minimal leaky degradation (in the absence of auxin) and reasonably rapid degradation upon addition of auxin (strain o401RS; Supplementary Table 2). Because the C-terminal tail of FPP synthase is important for the binding of substrate IPP in its catalytic pocket[39], Erg20p was then tagged with N-terminal Cup1-AID* through CRISPR/Cas9-mediated genome editing (strain o141; Supplementary Table 2). This resulted in NAA-dependent growth inhibition (Fig. 2b), whereas no inhibition was seen in the parent strain (o401RS; Supplementary Fig. 1). A limonene synthase and a GPP synthase (mutant Erg20p with an active site mutation that excludes FPP; *ERG20*[N127W]) were then introduced to both strain o401RS and o141 to deliver a full limonene production pathway. It should be noted that the *ERG20*[N127W] mutant can make a small amount of FPP[36]; this mutant was chosen to ensure that some flux to C15 derivatives still occurred in the Erg20p-tagged strain.

To compare two strains, it is important to ensure that mevalonate pathway flux into the precursor pool (IPP and DMAPP) is comparable. Some of our previous observations have suggested that induction of the *GAL* promoters can vary significantly between strains (unpublished data). We therefore examined *GAL* promoter induction in the engineered strains. We employed a Yellow Fluorescence-activating and Absorption-Shifting Tag (Y-FAST)[40], which requires the addition of fluorogen 4-hydroxy-3-methylbenzylidene rhodanine (HMBR) to fluoresce, to monitor the GAL promoter induction. This approach avoids interference with propidium iodide (PI) staining fluorescence when PI is used to examine cell viability using flow cytometry (Supplementary Table 6). Y-FAST was placed under the control of the *GAL10* promoter. Finally, the GAL80 transcriptional repressor (which represses *GAL* promoters in the absence of galactose) was knocked out to allow induction of *GAL* promoters upon depletion of glucose, and in the absence of galactose[38]. These manipulations produced strains LIM401 (untagged) and LIM141 (Erg20p tagged; conditional degradation; Fig. 2a).

LIM401 and LIM141 were characterised in dodecane-overlayed MES-buffered YNB-glucose medium. We first tested the performance of both strains with NAA added at 10 h to 1 mM final concentration. We presumed: (1) that either Erg20p activity over the remaining exponential phase, or residual sterol pools in the cell generated during growth prior to addition of NAA, were sufficient to sustain growth prior to entering stationary phase; (2) that the residual FPP synthase activity of the *ERG20*[N127W] mutant may sustain sufficient pathway flux to supplement degradation of native *ERG20*;[38] and (3) that triggering depletion of Erg20p in an early phase might prevent the accumulation of FPP after the induction of *GAL* promoters-controlled MVA pathway upon diauxic shift at 13–15 h.

Under these conditions (NAA added at 10 h), both strains showed similar growth patterns, including exponential growth normally observed in *S. cerevisiae*, and reaching stationary phase with $OD_{600}$ around 10 (Fig. 2c), significantly lower than wild-type yeast, which reaches stationary phase with $OD_{600}$ around 15 under the same glucose dose[32,33]. The Y-FAST fluorescence demonstrated that *GAL* promoter induction occurred around 12 h, as glucose becomes depleted (Fig. 2d). Strain LIM141 showed a lower-level induction of Y-FAST expression compared to the non-tagged strain LIM401 (not significant).

To understand carbon allocations between different nodes of the prenyl phosphate metabolism, it is necessary to examine several different terpenoids. In mevalonate pathway-augmented strains, excess prenyl phosphate precursors are dephosphorylated by non-specific hydrolases to produce terpene alcohols[34,41,42] (Fig. 2a). Therefore, concentrations of limonene (C10, the target monoterpene), geraniol (C10), farnesol (C15), and geranylgeraniol (C20) were measured at 48 and 96 h. The reference strain LIM401 (*ERG20* wildtype) produced ~3 mg L$^{-1}$ limonene at 96 h. Significant farnesol and geranylgeraniol were produced in LIM401 (Fig. 2g–h), reflecting augmentation of the MVA pathway flux (farnesol and geranulgeraniol are not produced in strains with native MVA pathway flux)[34].

Conditional degradation of Erg20p with NAA added at 10 h delivered a ~14-fold increase in limonene production, to ~41 mg L$^{-1}$ in LIM141. Limonene was produced primarily during the 48–96 h period. Production of geraniol was not detectable in LIM401, whereas in LIM141, geraniol was produced at ~23 mg L$^{-1}$. Significant geraniol was also produced during the initial 48 h of the experiment. In LIM141, a significant decrease in farnesol and geranylgeraniol were observed. The increased production of limonene and geraniol and decreased production of farnesol and geranylgeraniol showed that depleting Erg20p via auxin-inducible protein degradation redirected metabolic flux for improved monoterpene production.

For a better understanding of the auxin-mediated protein degradation tool in metabolic engineering, we further tested the LIM141 strains without NAA addition or with NAA added at 24 h to trigger Erg20p depletion. Adding NAA at 24 h did not dramatically change the terpene production profiles in LIM141, though it did lead to a slight (but not statistically significant) decrease in the induction level of Y-FAST (Fig. 2d), compared to the conditions without NAA addition. Unexpectedly, in the absence of NAA addition to trigger Erg20p depletion, LIM141 showed increased production of monoterpenes limonene and geraniol and decreased production of farnesol, compared to *ERG20*-reference strain LIM401 (Fig. 2e–g). This suggests that there auxin-independent flux redirection occurs in strain LIM141. It further indicates that there are unpredictable and non-specific perturbation effects which can result from the introduction of the auxin-inducible protein degradation system. Such perturbation effects might be caused by the decreased efficiency of Erg20p expression or catalytic function (wither of which might be a result of the degradation tag, which might influence transcription efficiency, translation efficiency, or catalytic function); or could also be due to non-specific auxin-independent degradation of Erg20p. The latter cannot be excluded because the basal degradation had not been fully eliminated in the current system (as evidenced by the yEGFP reporter experiments – see Fig. 1).

Strain LIM141 exhibited similar physiological features both in the absence of NAA and when NAA was added at 24 h, compared to when NAA was added at 10 h. The features included: (1) improved growth during the post-exponential phase (Fig. 2c); (2) production of limonene primarily in the first 48 h rather than in the period from 48 h to 96 h, with limonene titres being higher at 48 h than the 10 h experiment (Fig. 2e); and (3) prenyl alcohols were produced consistently during the first 48-hour period and during the period from 48 h to 96 h. Overall, we observed lower geraniol titres, higher farnesol titres, and higher geranylgeraniol titres (Fig. 2f–h). This indicates that it is important to constrain the flux towards FPP production during exponential growth (before the diauxic shift) to deliver improved monoterpene production. It also suggests the possibility that prenyl pyrophosphate products downstream of ERG20 (i.e. FPP or GGPP) can inhibit upstream pathways and/or limonene synthase through an

as-yet unknown feedback mechanism (presumably through competitive inhibition; but this is yet to be investigated).

**Metabolic modulation: depleting Hxk2p boosts terpene (nerolidol) production.** Synthesis of terpenoids through the mevalonate pathway is energy demanding; however, *S. cerevisiae* is Crabtree-positive: in the presence of oxygen and excess glucose, cells adopt fermentative metabolism, converting glucose to ethanol and repressing energy-production pathways[43,44]. This pathway produces less cellular energy than respiration. After glucose depletion, metabolism is reprogrammed to ethanol-consuming respiratory metabolism.

Glucose and other fermentative sugars are the current primary carbon sources in industrial applications. Production of terpenes in yeast is commonly achieved by placing terpene anabolic genes under control of the strong galactose-inducible *GAL* promoters[32,38], which are de-repressed in the absence of glucose; however, in these strains terpene production is less efficient under glucose fed-batch cultivation than that under ethanol fed-batch cultivation[45]. We hypothesise that this might be due to the glucose-mediated Crabtree effect and/or glucose-dependent repression of *GAL* promoters driving the terpene anabolic pathway.

Here, we applied the auxin-mediated protein degradation mechanism to perturb the glucose signalling pathway, with the aim of derepressing *GAL* promoters and respiratory metabolism in the presence of glucose. The regulatory target we chose was hexokinase 2 (Hxk2p). This enzyme has two established functions: (1) it functions as predominant hexokinase on glucose, allowing entry of fructose and glucose into central carbon metabolism (Fig. 3a). Expression of other hexokinase genes *HXK1, GLK1,* and *EMI2* is subjective to glucose repression[46,47]. (2) It also functions in glucose signalling and transcriptional regulation (Fig. 3b). In the glucose signalling pathway, Hxk2p interacts with the Snf1p complex, which blocks Mig1p phosphorylation under high-glucose conditions[48] and stimulates Reg1p phosphorylation under low-glucose conditions[49]. Phosphorylation of Mig1p is essential for de-repression of glucose-repressible genes[48], including *GAL* genes. Previous physiological studies have shown that *HXK2* deletion increased respiration and biomass production on glucose[50,51].

For auxin-mediated depletion of Hxk2p, we fused the AID*-Cup1p tag to the Hxk2p C-terminus through CRISPR/Cas9-mediated genome editing (strain o138; Supplementary Table 2). The C-terminus was chosen because the N-terminal residues of Hxk2p are important for its role in glucose signalling[52–54] and it was critical not to interfere with this function. The C-terminal of Hxk2p is important for catalytic activity[54].

Monoterpene production in yeast is inefficient due to poor catalytic activity of monoterpene synthases, resulting in conversion of excess GPP to geraniol in limonene producing strains[33] (Fig. 2f). Accumulation of prenyl pyrophosphates is suspected to exert a toxic effect on cells[55]. These adverse effects might impede evaluation of the full potential of metabolic engineering strategies. We therefore used yeast strains engineered for the synthesis of the sesquiterpene nerolidol, for which we have achieved substantial product titres[32,33], indicating a more efficient flux through to the target product. Through 2A sequence-mediated polycistronic expression[56] of Y-FAST and *Actinidia chinensis* (kiwifruit) nerolidol synthase (*Y-FAST-2A-AcNES1*), nerolidol titre was improved to ~1.7 g L$^{-1}$ in flask cultivation with 20 g L$^{-1}$ glucose as the carbon source (Supplementary Result 1 and Supplementary Fig. 2: *HXK2* wild-type strain NLD401). We used Y-FAST as a proxy for translation of the *GAL* promoter-driven *AcNES1* transcripts in the polycistronic construct. Induction of the *GAL* promoter-controlled *Y-FAST-2A-AcNES1* (as inferred from the Y-FAST fluorescence) and nerolidol production in the reference *HXK2* strain NLD401 was no different in the presence or absence of NAA (Fig. 3f, g and Supplementary Fig. 2).

To evaluate the effect of Hxk2p depletion on nerolidol production, an *HXK2-AID*-CUP1* nerolidol-producing strain NLD128 was developed (Supplementary Result 2 and Supplementary Fig. 3). However, there were two types of NLD128 clones: one type showing higher Y-FAST expression and higher nerolidol production than another type (Supplementary Result 2 and Supplementary Fig. 3). *HXK2* modifications in both types of clones was confirmed by DNA sequencing, and the sequence was as being designed. The causes of this variation might be due to the variations in plasmid copy number or at other genetic loci. In the following study, we chose a high-nerolidol-prduction clone NLD128-1 to further examine the effects of Hxk2p depletion. In the absence of NAA, the *HXK2-AID*-CUP1* strain NLD128-1 had a faster exponential growth rate and higher overall biomass yield compared to its reference strain NLD401 (Fig. 3c–d and Supplementary Fig. 2c, d, f). This was accompanied by faster carbon uptake, both on glucose and ethanol, (Fig. 3e, h) compared to NLD401 with NAA added. Carbon uptake in this strain was also faster than in previous nerolidol producing strains and in wild-type *S. cerevisiae* CEN.PK strains[26,32]. Acetate accumulation was significantly reduced (Fig. 3i), and production of mevalonate, a key intermediate in the mevalonate pathway, increased (Fig. 3l).

Y-FAST fluorescence in strain NLD128-1 was already observed at ~9-fold of the levels in strain NLD401 during the exponential growth on glucose (Fig. 3f), indicating very high translation in the polycistronic construct. Y-FAST was further induced after glucose depletion but remained constant after 24 h, whereas continuous induction was observed in strain NLD401. Although NLD128-1 exhibited a maximal *Y-FAST* level of half of that in NLD401 (Fig. 3f and Supplementary Fig. 4), suggesting weaker induction of *GAL* promoters and lower expression levels of the nerolidol synthase, NLD128-1 produced almost 3 g L$^{-1}$ nerolidol at 72 h, a ~70% increase compared with NLD401 (Fig. 3g). These data demonstrate that the *AID*-CUP1-HXK2* modification significantly changed both induction of the *GAL* promoter and carbon metabolism in nerolidol-producing strain. A LC-MS/MS-based proteome assay showed 38% decrease of Hxk2p abundance in NLD128-1, compared to strain NLD401 (Supplementary Fig. 5a), potentially indicating auxin-independent decrease of tagged Hxk2p although the auxin-inducible protein degradation tool with reduced non-auxin-dependent degradation was used.

To trigger Hxk2p depletion, NAA was added to both pre-culture and the main culture of NLD128-1 at inoculation. Hxk2p depletion was confirmed through a LC-MS/MS-based proteome assay (Supplementary Fig. 5b). Degradation of Hxk2p resulted in severe inhibition of glucose uptake and growth (Fig. 3c–e). Y-FAST expression was de-repressed in the early exponential phase (Fig. 3f). We also observed that the Y-FAST induction in strain NLD128-1 was a gradual induction pattern rather than the binary induction pattern seen in strain NLD401 (Supplementary Result 3 and Supplementary Fig. 4). Consistent with the physiological features in *hxk2Δ* cells with increased respiration[51], glycerol and acetate production was reduced (Fig. 3i, k). Despite the severe reduction in glucose uptake and growth rate on glucose, a nerolidol titre of ~3.4 g L$^{-1}$ was reached (Fig. 3g). The specific nerolidol production rate was higher in the exponential phase when glucose was the carbon source than that in the ethanol-growth phase (Fig. 3j). This demonstrated that removing Hxk2p function can lift glucose repression on nerolidol production.

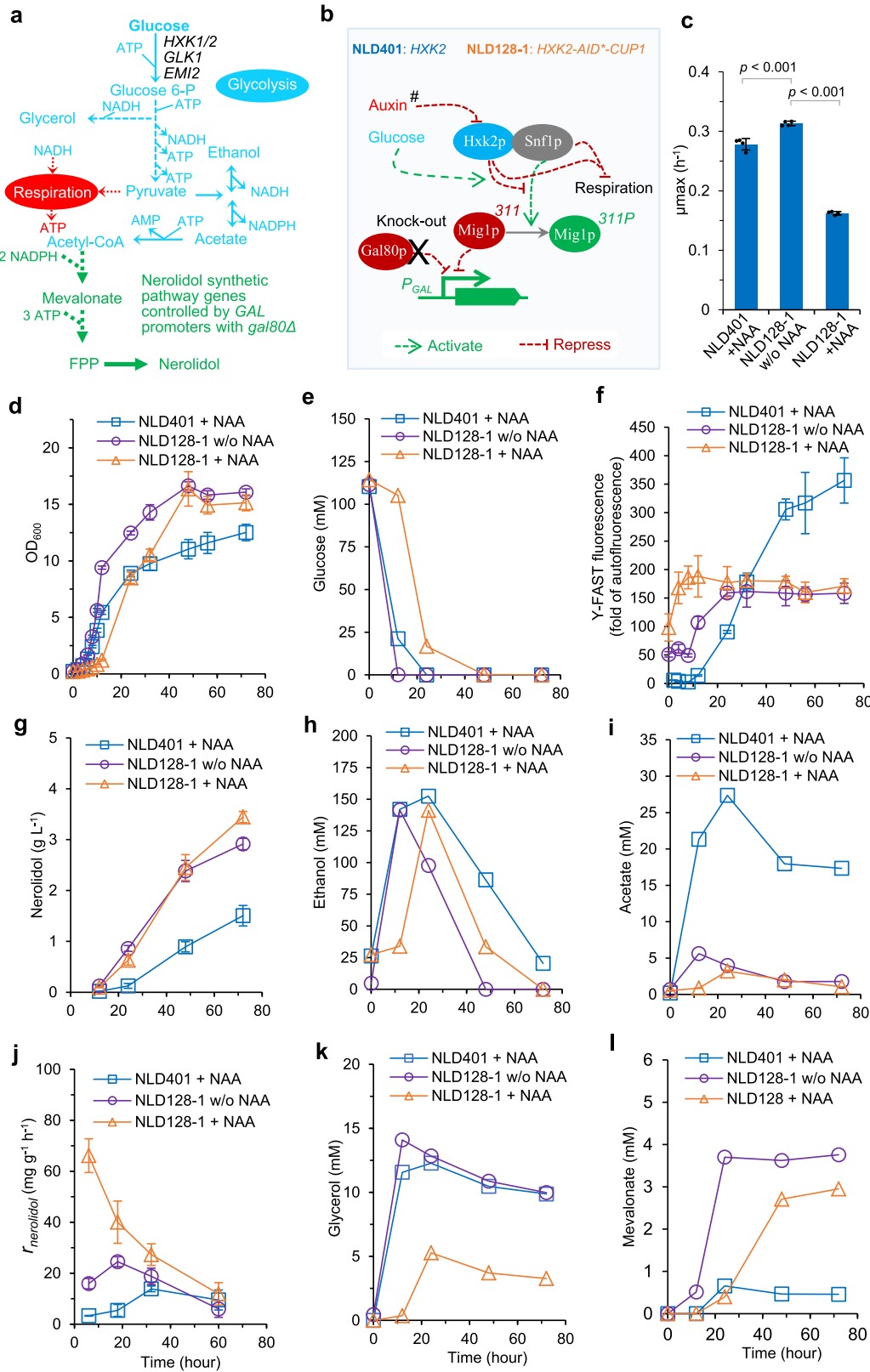

**Growth arrest: depletion of Acc1p arrests growth but not production.** Heterotrophic microbes like *S. cerevisiae* exhibit co-regulation of cell growth and metabolism according to nutrient availability for maximum cell proliferation[57], resulting in distinct growth phases[32]. Limiting biomass production while maintaining active metabolism may allow channelling of more carbon flux into the desired product[58] – thereby contributing to a continuous stable production phase. Here, we employed the auxin-mediated gene inactivation tool to test an engineering strategy to decouple cell growth and metabolism.

Regulating lipogenesis is an essential metabolic mechanism to control the proliferation of certain cells[59,60]. Limiting lipogenesis

**Fig. 3 Engineering hexokinase 2 (Hxk2p) modulated glucose metabolism and boosted nerolidol production. a** Schematic of metabolic pathways for nerolidol synthesis. Hexokinase *HXK1/2*, Glucokinase *GLK1/EMI2*, FPP farnesyl pyrophosphate synthase. **b** Schematic of Hxk2p/Snf1p-mediated glucose repression on respiration and on *GAL* promoters via Mig1p. #, In *HXK2-AID\*-CUP1* strain NLD128-1, auxin mediates Hxk2p depletion, and in *HXK2*-wildtype strain NLD401 Hxk2p is not regulated by auxin. **c–l** Characterisation of strain NLD401 and strain NLD128 with or without addition of 1 mM 1-Naphthaleneacetic acid (NAA) at 0 h and in precultures. Two-phase flask cultivation on 20 g L$^{-1}$ glucose was employed. Y-FAST was under the control of the *GAL2* promoter. Y-FAST fluorescence was measured in cells mixed with 20 μM 4-hydroxy-3-methylbenzylidene rhodanine (HMBR). Y-FAST fluorescence is expressed as the fold-change of exponential-phase auto-fluorescence of a reference strain GH4. The growth, Y-FAST fluorescence, and nerolidol production in strain NLD401 without NAA addition is available in Supplementary Fig. 2. Mean values ± standard deviations are shown in **c, d, f, g, j** (N = 4 independent biological replicates for NLD401 and independent cultures for NLD128-1). Mean values are shown in **e, h, i, k, l** (N = 2 independent biological replicates for NLD401 and independent cultures for NLD128-1). Two-tailed Welch's t-test was used for comparing two groups in **c**. Source data in **c–l** are provided as a Source Data file.

by diversion of acetyl-CoA from fatty acid metabolism and into terpenoid production may be an effective mechanism to both impair cell growth and increase terpenoid biosynthesis (Fig. 4a). We selected the essential lipogenic enzyme acetyl-coA carboxylase Acc1p as the depletion target. For auxin-mediated depletion of Acc1p, we fused the Cup1p-AID* tag to the Acc1p N-terminus through CRISPR/Cas9-mediated genome editing (strain o138; Supplementary Table 2). We targeted the N-terminal because the N-terminal residues of Acc1p are exposed outside and the C-terminal residues are in a pocket[61].

Fusing Cup1p-AID* to the N-terminus of Acc1p (*CUP1-AID\*-ACC1*, *GAL80*-wildtype, without nerolidol synthetic module; strain o138) resulted in impaired growth in the presence of NAA (Fig. 4b). This modification was used to generate a nerolidol production strain, NLD138 (*CUP1-AID\*-ACC1*; *GAL* promoter-controlled MVA pathway, *ERG20*, and *Y-FAST-2A-AcNES1*; and *gal80Δ*).

In flask cultivation with 20 g L$^{-1}$ glucose as the carbon source, no difference was observed between the reference strain NLD401 (wild-type *ACC1*) and strain NLD138 during the exponential growth phase (Fig. 4c and Supplementary Fig. 2c). In the absence of NAA, strain NLD138 showed a decreased growth, slightly faster Y-YAST induction during the post-exponential growth phase, and produced similar amount of nerolidol at 72 h, compared to strain NLD401 (Fig. 4c–e, Supplementary Fig. 2c, and Supplementary Fig. 2f–g). These changes demonstrate an auxin-independent perturbation in the *CUP1-AID\*-ACC1* strain.

When NAA was added at 10 h to trigger Acc1p depletion, strain NLD138 showed growth arrest after 24 h (Fig. 4c). Y-FAST fluorescence plateaued after 48 h, suggesting that induction stopped at 48 h (Fig. 4d). After growth arrest, cells were still metabolically active. The nerolidol titre at 72 h was ~0.76 g L$^{-1}$, 43% of the titre in the absence of NAA addition. Consistent with this, the specific nerolidol production rate was induced to lower levels compared to the reference strain NLD401 between 24 h and 72 h (Fig. 4f). This might result from stalling *GAL* promoter induction (the pathway genes for nerolidol synthesis are controlled by *GAL* promoters).

NAA addition at 24 h led to growth arrest in strain NLD138 after 48 h, while strain NLD401 continued to grow (similar to the growth profile in the absence of NAA; Fig. 4c and Supplementary Fig. 2c). Depleting Acc1p after 24 h did not significantly influence Y-FAST induction (Fig. 4d), indicating that *GAL* promoter-driven expression was relatively unperturbed; however, nerolidol production was significantly improved, with the titre increased by ~36% to ~ 2.2 g L$^{-1}$ (Fig. 4e). The specific nerolidol production rate was the same during the first 24 h, but increased by ~50% during the period from 24 h to 72 h, compared to strain NLD401 (Fig. 4f). This indicates that the growth-arrested cells maintained a superior metabolic state for nerolidol production when Acc1p depletion was triggered after the diauxic shift (with the induction of *GAL* promoters unperturbed).

For a better understanding of the effects of growth arrest in NLD138 when the *GAL* promoters were fully de-repressed, we further characterised strain NLD138 and the reference strain NLD401 with ethanol as the sole carbon source, and with NAA added at 0 h. Neither of the strains showed exponential growth (Fig. 4g), possibly as a result of metabolic imbalance due to the strong induction of *GAL* promoter-controlled mevalonate pathway (as shown by the induction of Y-FAST fluorescence; Fig. 4h). However, the growth of strain NLD138 was much more severely impaired. Induction of Y-FAST halted after 24 h in strain NLD138, whereas it continued in strain NLD401. Although growth was severely impaired, strain NLD138 cells were metabolically active, as shown by nerolidol production (Fig. 4i). The specific nerolidol production rates in strain NLD138 were similar to the rates in strain NLD401. These results indicate that when cells are grown on ethanol with nerolidol synthetic pathways (including the mevalonate pathway) induced to a high level, Acc1p depletion did not affect acetyl-CoA flux redirection. The specific nerolidol production rates in growth-arrested NLD138 cells did not decline dramatically after the more prolonged incubation (Fig. 4j), and showed survival rates similar to the reference without growth arrest (Supplementary Table 6).

These results demonstrate that growth-arrested cells from auxin-inducible depletion of Acc1p are metabolically active, and that nerolidol productivities in the growth-arrested cells correlate with the induction state of nerolidol synthetic pathways.

## Discussion
Loss-of-function perturbation triggering cellular responses at genetic, metabolic, and physiological levels is an important strategy in metabolic engineering for improved production of target products. Auxin-inducible protein degradation has been recognised as a powerful tool to remove target proteins by adding a small molecule inducer. Here, we engineered this mechanism and exemplified its application in metabolic engineering for terpene production in yeast. We also investigated cellular responses to depletion of metabolic proteins.

An auxin-inducible protein degradation mechanism with reduced basal (leaky) degradation was reconstructed and optimised. The optimal circuit comprised a mini-AID* fused with Cup1p and a Skp1p-TIR1 fusion (Fig. 1). This system showed a compromise between an effective degradation rate and decreased auxin-independent basal degradation. However, in our work, tagging Erg20p and Hxk2p with AID tag still led to dramatic effects in terpene-producing yeast when auxin was not added. In *CUP1-AID\*-ERG20* strain, such tagging alone could resulted in flux redirection to improve limonene production, although further improvement required NAA addition before diauxic shift (Fig. 2). In *HXK2-AID\*-CUP1* strain, the *GAL* promoter was de-repressed, nerolidol production was high, glucose consumption rate increased, and acetate production decreased (Fig. 3). It is likely that the quantum of effect if variable across target proteins

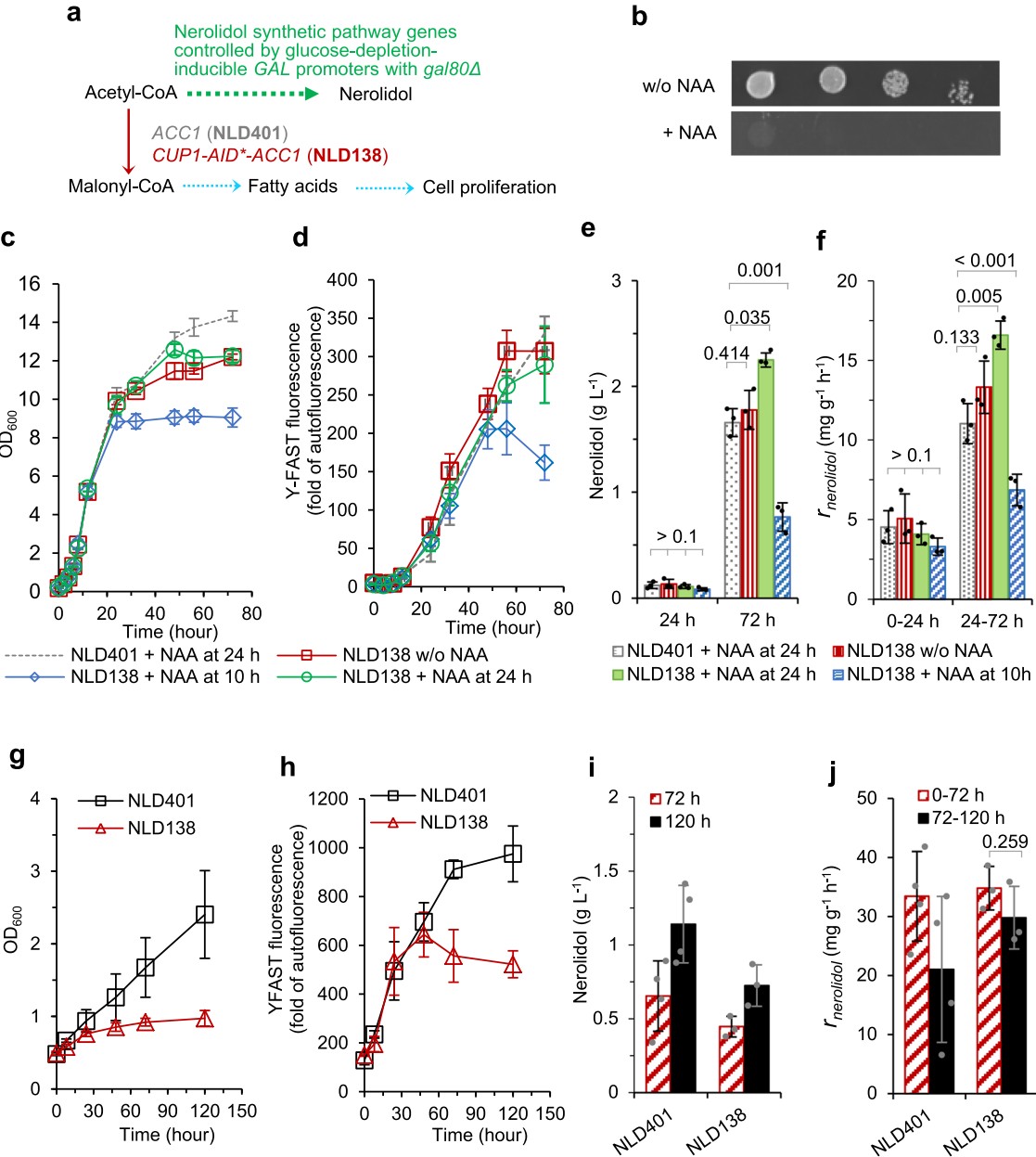

**Fig. 4 Depleting malonyl-CoA synthase Acc1p arresting growth but not compromising terpene production. a** Schematic of pathway engineering in strain NLD401 (*ACC1* wildtype) and strain NLD138 (*CUP1*-AID*-*ACC1*). **b** Growth of the background strain o138 (*CUP1*-AID*-*ACC1*, *GAL80*, without nerolidol synthetic module) on YNB-glucose agar with or without the addition of 1 mM 1-Naphthaleneacetic acid (NAA). The agar plate was incubated at 30 °C for 48 h. **c–f** Characterisation of strain NLD401 and strain NLD138 without NAA addition or with 1 mM NAA added at 10 h or 24 h in two-phase flask cultivation on 20 g L$^{-1}$ glucose. The growth, Y-FAST fluorescence, and nerolidol production in strain NLD401 without NAA addition is available in Supplementary Fig. 2. **g–j** Characterisation of strain NLD401 and strain NLD138 with 1 mM NAA added at 0 h in two-phase flask cultivation on 2% (v/v) ethanol. Y-FAST was under the control of the *GAL2* promoter. Y-FAST fluorescence was measured in cells mixed with 20 μM 4-hydroxy-3-methylbenzylidene rhodanine (HMBR). Y-FAST fluorescence is expressed as fold-change of exponential-phase auto-fluorescence of a reference strain GH4. Mean values ± standard deviations are shown (*N* = 3 or 4 independent biological replicates). Two-tailed Welch's *t*-test was used for comparing two groups, and *p* values are shown above the bars. Source data in **c–j** are provided as a Source Data file.

and under differing conditions due to idiosyncrasies of the test systems. Therefore, it is advisable to users to test each protein. We analysed Hxk2p abundance, which showed a decrease of Hxk2p (Supplementary Fig. 5a). We postulate that these broad-scale metabolic responses resulted from basal decrease of protein abundance of Erg20p and Hxk2p and/or negative effects on normal protein function upon modification with AID tag, like that modification on Hxk2p may influence its function in glucose sensing networks[62]. These results suggest that modifying a target

protein by tagging may lead to a dramatic regulatory effect in some cases.

It is necessary to emphasise that, in the three examples (Erg20p, Hxk2p, and Acc1p) we tested, tagging target protein with the AID tag had idiosyncratic effects for individual proteins, making prediction of the phenotype challenging. Despite this, tagging did not dramatically influence growth until auxin NAA was added to trigger the depletion of target protein. Depletion of these three proteins not only lead to improved production of the

target terpene products, but also caused broad phenotypic responses in yeast cells, including responses in metabolism, gene expression, and cell growth (Figs. 2–4). We also found that timing of induction for protein depletion is a critical consideration for the process. For example, triggering Erg20p depletion in the exponential phase (at 10 h) resulted in significant flux redirection toward limonene production (Fig. 2). We presume that this is due to FPP accumulation prior to depletion of Erg20p (which would result in negative regulation of the upstream pathways, decreasing flux to FPP). However, triggering Erg20p depletion in the post-exponential phase (at 24 h) was less successful, presumably because FPP accumulation prior to depletion of Erg20p inhibited the upstream pathways. Triggering Acc1p depletion at 10 h and at 24 h resulted in growth-arrested cells with varying nerolidol productivities, which positively correlated with induction levels of *GAL* promoters (Fig. 4). These results demonstrate that the auxin-inducible protein degradation can be applied for a swift perturbation, and that removing the target protein at different stages delivers an improved understanding of the regulation of metabolism and other intracellular processes. This could be performed in combination with model-driven system biology studies[63–65] to further examine metabolism.

In particular, we characterised that depletion of the GPP-consuming enzyme Erg20p redistributed metabolic flux towards monoterpenes (Fig. 2), demonstrating the application of auxin-inducible protein degradation in metabolic engineering. However, the auxin-inducible degradation tag strategy for Erg20p delivered a lower titre of limonene than our previous approach of constitutive degradation of Erg20p mediated by an N-terminal degron[33] (43 mg/L vs 76 mg/L). These differences could be due to a combination of factors, including variations in degradation efficiency and transcriptional efficiency, which may affect final Erg20p amount and consequently result in unpredictable system effects when modifying this essential gene, and other less obvious variations, all have an influence on limonene titre.

Here, we examined terpene alcohols (Fig. 2), which are produced when prenyl phosphates accumulate by the action of nonspecific phosphorylases[34] and presumably act as a proxy for intracellular prenyl phosphate levels. In *E. coli*, FPP accumulation led to a complex stress effect[55]. It is not clear whether such toxicity also exist in yeast. In strains overexpressing the MVA pathway in the absence of a sink, this post-exponential phase growth impairment also occured, and we speculated that this might be caused by a proposed toxicity effect from over-accumulation of FPP in the presence of increased upstream pathway flux[33]. In the same scenario, the growth impairment was obersed in *ERG20*-wildtype strain LIM401 (Fig. 2). However, triggering Erg20p depletion at 10 h before cells shifted into ethanol growth phase resulted in growth impairment during the post-exponential phase in flask cultivation (Fig. 2c). Post-exponential growth impairment was significant in the same strain with the depletion triggered in ethanol growth phase (Fig. 2c) as well as in previous strains with the constitutive N-terminal degron approach in flask culture[33]. However, the lack of growth defect in these cases, where prenyl alcohols also accumulate[33], suggests that prenyl phosphate accumulation not be linked to the post-exponential growth defect. It is therefore unclear exactly what causes the post-exponential phase toxicity effect. We speculate that it may be a cellular imbalance in prenyl phosphate metabolism, which requires the examination of this phenomenon in future work.

Acc1p depletion caused growth arrest, but nerolidol productivity was not diminished (Fig. 4). This demonstrates successful decoupling of growth and nerolidol production in this strain. However, the productivities of growth-arrested cells correlated with induction state of nerolidol synthetic pathways

controlled by *GAL* promoters. Flux redirection by Acc1p depletion was only observed under specific conditions when heterologous nerolidol synthesis was induced to a medium level in ethanol-growth phase in flask cultivation on glucose (Fig. 4f) rather than when nerolidol synthesis was induced to doubled levels in flask cultivation on ethanol (Fig. 4j). Restricting biomass production by limiting flux through an essential pathway has previously been successful in redirecting flux towards production of interested heterologous metabolites in different organisms. Examples include improving production of myo-inositol derivatives by restricting glycolysis[66], improving shikimate production by restricting the downstream shikimate pathway essential for aromatic amino acid synthesis[66], improving fatty acid production by restricting leucine synthesis or nitrogen starvation[67], and improving carbohydrate production by restricting the tricarboxylic acid cycle[68]. In our case, we presumed that a decrease in Acc1p activity presumably increased acetyl-CoA availability for redirection into the mevalonate pathway, leading to improved nerolidol titre and improved specific production rate in flask cultivation on glucose. The observed effect of improved nerolidol production was modest and required specific conditions.

In metabolic engineering, unpredictable off-target effects from gene and protein modification can occur. We were interested in transcriptional behaviour of the *GAL* promoters in the context of various modifications we made, because we used them to drive terpene anabolic pathway genes in terpene-producing strains and variations in promoter activity would have a significant effect on interpretation of results. Using Y-FAST as a transcriptional reporter in our auxin-inducible degradation strains, we observed dramatic, inconsistent, and unpredictable variations in *GAL* promoter behaviour between different strains. In nerolidol-producing strains, depletion of Hxk2p or Acc1p influenced the induction strength of the *GAL2* promoter (Figs. 3f, 4h), whereas engineering Hxk2p also altered the *GAL*-promoter induction pattern in yeast populations (Supplementary Fig. 4; Supplementary Result 3). In the reference strains without modification of Hxk2p, yeast cells showed a binary induction of the *GAL* promoter, while a large proportion of yeast cells did not show induced Y-FAST fluorescence in the post-exponential phase (Supplementary Fig. 4d). Modification and depletion of Hxk2p resulted in a homogeneous yeast population with induced Y-FAST expression from the *GAL* promoter in the exponential phase and post-exponential phase (Supplementary Fig. 4f, g). We suspect that changes in *GAL* promoter induction patterns contribute to changes in production profiles in different engineering approaches. Improved *GAL* promoter induction homogeneity might contribute to the improved nerolidol production in Hxk2p-engineered strain.

In summary, by developing strategies for flux redirection, metabolic modulation, and growth arrest, we showed that auxin-mediated protein degradation can be broadly applied in metabolic engineering for investigation of metabolic optimisation strategies. The approach provides a rapid, flexible loss-of-function perturbation switch. The method can also be used in systems biology studies for a better understanding of metabolism and metabolic regulation. Application of this tool in an industrial setting will require further examination to determine feasibility. Here, we have demonstrated its use for rapid phenotyping of metabolic perturbations, which might be useful in a metabolic engineering context. Different products present different production challenges, and the ultimate choice of engineering approaches is dependent on the metabolic context of the desired product. The utility of the tools we have developed will need to be tested individually for each metabolic engineering project. They will have the most utility in cases where inducible downregulation at the protein level is desirable – e.g. for very stable proteins, for

proteins in essential pathways, and for decoupling production from metabolism. Further optimisation of auxin-inducible protein degradation tools might be achieved using other systems with a lower basal degradation and rapid depletion of target protein upon auxin addition. Similar systems have been reported recently in mammalian cells[20,25]. In addition, integrating with auxin-mediated quorum-sensing mechanisms[22,69] may trigger automatic regulation in metabolic engineering. Implementing Y-FAST as a tool, we observed single-cell level changes in the induction of the *GAL* promoter in terpene-producing strains with auxin-inducible degradation of an enzyme, which can instruct further studies to understand metabolic and genetic regulation in yeast.

## Methods

**Plasmid and strain construction**. Plasmids used in this work are listed in Supplementary Table 1, and strains are listed in Supplementary Table 2. Primers used in polymerase chain reaction (PCR) and PCR performed in this work are listed in Supplementary Table 3. Plasmid construction processes are listed in Supplementary Table 4. Yeast strain construction processes are listed in Supplementary Table 5. Molecular Cloning Designer Simulator[70] was used to design the plasmids used in this study.

**Two-phase flask cultivation**. Terpene-producing strains were characterised using dodecane-overlayed two-phase flask cultivation. The medium contained $6.9 \, \text{g L}^{-1}$ YNB (FORMEDIUM#CYN0402) with $20 \, \text{g L}^{-1}$ glucose or 2% (v/v) ethanol as the carbon source. MES (100 mM) was used as buffer with the pH adjusted to 6.0 with ammonium hydroxide. Dodecane was used as a non-toxic organic phase for the in-situ extraction of the terpene products. Strains were recovered from the glycerol stock on YNB agar containing $200 \, \text{g L}^{-1}$ glucose[71]. Before initiating the two-phase flask cultivation, cells were pre-cultured in MES-buffered YNB-$20 \, \text{g L}^{-1}$ glucose to exponential phase (OD600 between 1 and 4) and collected by centrifugation. Collected cells were then resuspended in fresh fermentation medium. To initiate the cultivation, appropriate volumes of pre-cultured cells were transferred to MES-buffered YNB medium with $20 \, \text{g L}^{-1}$ glucose to an initial OD600 of 0.2 in a total volume of 23 mL medium in a 250 mL flask, and 2 mL sterile dodecane was added after inoculation. For monoterpene-producing strains, flasks with a screw-cap were used to avoid product loss by evaporation. NAA (500 mM, dissolved in ethanol) was added at specific time points to a concentration of 1 mM to trigger auxin-inducible protein degradation. Cell cultivations were performed at 200 rpm and 30 °C. In the first 12 h of cultivation, 3 mL culture was sampled, at later time points 50 μL dodecane and 500 μL culture were sampled. The samples were immediately used for cell density analysis and/or fluorescence-based analysis. Dodecane samples and supernatant of centrifuged culture were stored at −80 °C for extracellular metabolite analysis.

**Drop plate assay**. Cells grown in rich medium containing $20 \, \text{g L}^{-1}$ peptone, $10 \, \text{g L}^{-1}$ yeast extract, and $20 \, \text{g L}^{-1}$ glucose were washed and resuspended in water, and was incubated at 30 °C for 6 h to prepare quiescent cells. Quiescent cells were diluted in water to prepare four samples with OD600 equalling to 1, 0.1, 0.01, and 0.001, separately for each strain. Diluted samples (4 μl) were dropped on YNB $20 \, \text{g L}^{-1}$ glucose agar with or without 1 mM NAA added. The plates were incubated for 48 h at 30 °C before imaging.

**Flow cytometry analysis**. Fluorescence in single cells was analysed using a BD Accuri™ C6 flow cytometer (BD Biosciences, USA). For analysis of auxin-inducible protein degradation with yEGFP as the fluorescent reporter protein or the analysis of *Y-FAST-2A-yEGFP*-expressing cells, cells were grown in MES-buffered YNB to early exponential phase (OD600 from 0.7 to 1.5) and directly analysed. For analysis of induction of *GAL* promoters, cells sampled from two-phase flask cultivation were analysed. For analysis of Y-FAST fluorescence, 100-time-concentrated HMBR, synthesised as reported previously[40] and dissolved in dimethyl sulfoxide, was added to the samples to 20 μM final concentration and the sample was mixed before analysis. FSC.H threshold was set at the value of 250,000 for exclusion of debris particles. GFP and/or Y-FAST fluorescence was excited by a 488 nm laser and monitored through a 530/20 nm bandpass filter (FL1.A), with 10,000 events recorded per sample. Mean values of FSC.A, SSC.A, and FL1.A for all detected events were extracted using a BD Csampler software (BD Accuri C6 software version 1.0.264.21). GFP or Y-FAST fluorescence level was expressed as the percentage of the average background auto-fluorescence from the exponential-phase cells of GFP-negative reference strain GH4 as described previously[26].

**Metabolite analysis**. The Metabolomics Australia Queensland Node analysed extracellular metabolites. Dodecane samples (in some cases, diluted with dodecane) were diluted in 40-fold volume of ethanol. The ethanol-diluted samples (20 μL) were injected. A Zorbax Extend C18 column (4.6 × 150 mm, 3.5 μm, Agilent PN: 763953-902) equipped with a guard column (SecurityGuard Gemini C18, Phenomenex PN: AJO-7597) was used. Analytes were eluted at 35 °C at 0.9 mL/min using the mixture

of solvent A (water) and solvent B (45% acetonitrile, 45% methanol, and 10% water), with a linear gradient of 5–100% solvent B from 0–24 min, then 100% from 24–30 min, and finally 5% from 30.1–35 min. Analytes of interest were monitored using a diode array detector (Agilent DAD SL, G1315C) at 202 nm wavelength. Analytical standards of geraniol (98% purity; Sigma-Aldrich #163333), linalool (97 % purity; Sigma-Aldrich #L2602), *trans,trans*-farnesol (96% purity; Sigma-Aldrich #277541), trans-nerolidol (93.7% purity; Sigma-Aldrich #04610590), limonene (Sigma-Aldrich #W263303), and geranylgeraniol (85% purity; Sigma-Aldrich #G3278), were used to prepare the standard curve for quantification.

Glucose, ethanol, acetate, glycerol, and mevalonate were analysed through ion-exclusion chromatography[72]. Ion-exclusion chromatography was performed using an Agilent 1200 HPLC system and an Agilent Hiplex H column (300 × 7.7 mm, PL1170–6830) with guard column (SecurityGuard Carbo-H, Phenomenex PN: AJO-4490). Analytes were eluted isocratically with 4 mM $H_2SO_4$ at 0.6 mL min$^{-1}$ at 65 °C. Glucose, ethanol, glycerol, and mevalonate were monitored using a refractive index detector (Agilent RID, G1362A); acetate and mevalonate were detected using an ultraviolet-visible light absorbance detector (Agilent MWD, G1365B) at 210 nm.

**Physiological feature calculation**. Physiological parameters were calculated as reported previously[73]: the maximum growth rates are the linear regression coefficients of the ln OD600 versus time during the exponential growth phase; one unit of OD600 equals $0.23 \, \text{g L}^{-1}$ biomass; the specific production rate of nerolidol (mg g$^{-1}$ h$^{-1}$) was calculated by dividing the increase in nerolidol titre (mg L$^{-1}$) with the integral area (g h L$^{-1}$) of biomass over a defined time period[73].

**Reporting summary**. Further information on research design is available in the Nature Research Reporting Summary linked to this article.

## Data availability

Source data are provided with this paper. Plasmids used in this study are available on request or on Addgene. Any other relevant data are available from the authors upon reasonable request.

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

## Acknowledgements

B.P. and this research are supported by a CSIRO synthetic biology future science fellowship and the University of Queensland. Metabolite analysis was performed by Dr. Manual Plan and LC-MS/MS-based proteome assay was performed by Dr. Gert Tablo and Dr. Lian Liu in the Metabolomics Australia (Bioplatform Australia) Queensland Node. Yeast strains in this study derives from CEN. P.K. background strains, which were provided by EUROSCARF (Scientific Research and Development GmbH, Germany) under a non-commercial licence.

## Author contributions

B.P., C.E.V., and G.D. contributed to the conception of the project. B.P. and Z.L. designed experiments. Z.L. and B.P. performed the experiments. B.P. and Z.L. drafted manuscript. C.E.V., B.E.E., and G.D. revised manuscripts. C.E.V., G.D., and B.E.E. participated in the support and coordination of the project. All authors contributed to result analysis and discussion.

## Competing interests

The authors declare no competing interests.
