## [Peer Review File · Nature Communications]

Reviewers' Comments:

Reviewer #1:

Remarks to the Author:

NCOMMS-20-31830

An auxin-mediated protein depletion switch for metabolic perturbation in yeast

Zeyu Lu, Bingyin Peng, Birgitta E. Ebert, Geoff Dumsday, Claudia E. Vickers.

Saccharomyces cerevisiae is an important platform strain for the production of various biofuels, biochemicals, and biopharmaceuticals. A large variety of industrially useful compounds are part of the isoprenoid family and use the same initial pathway to produce the requisite precursors. In recent times the synthetic biological approach of metabolic pathway engineering has become the method of choice for optimised production of selected products. Part of this approach is the re-direction of metabolic flux towards the desired pathway. Deletion of genes for enzymes that would direct precursors towards alternative metabolic routes is commonly used, but problematic when absence of the enzyme is detrimental to growth or survival of the yeast, while conditional deletion can be slow and ineffective depending on the protein.

Here the authors demonstrate that the use of the well-established Auxin inducible degron system can provide an effective alternative. The AID has become a favoured inducible protein degradation system due to its fast and efficient inducibility by the orthologous plant hormone auxin or its analogues, even though some constitutive leakage has been observed in several studies.

This is a well written manuscript describing several well-designed experiments, with a few reservations as mentioned below. Together the findings of this manuscript achieve to highlight the promising potential of the auxin inducible degron system for metabolic engineering and show it can be smart solution to diverse aspects such as flux redirection and the decoupling of growth and production. Interestingly the authors have used a different title for the supplementary information file which in my opinion provides a better coverage of the contents of the manuscript.

In the first part of this study the authors set up the AID system for their yeast strains and play around with a few construct formats to find the optimal set-up based on loss of fluorescence of AID tagged yEGFP fusion proteins. Interestingly fusion with CUP1 appears to prevent premature loss of fluorescence, while three other fusion partners make things worse. Whether this is due to increased basal protein level/stability or enhanced solubility remains unclear and whether CUP1 stabilises the AID-tag moiety, yEGFP protein or the combination of those. More importantly I wonder if the CUP1 effect is a general improvement of the AID-tag or, as would seem likely, that the effectiveness of the fusion is particular to each specific protein. Therefore, the question arises whether the CUP1-AID fusion is equally effective on the specific metabolic enzymes tagged in this study. It would be informative to see comparative profile of the abundance of the various tagged and untagged proteins over time in the presence or absence of inducer by Western blot.

In the next sections the authors apply the AID system to three different enzymes representing three aspects of metabolic engineering, building on their knowledge and in-house selection of engineered yeast strains and constructs, and appear to achieve improvements in each case.

Below are a few, mostly minor comments on these sections:

Firstly, despite their efforts to establish the optimal format for their AID set-up in Fig1, they appear to use a mish-mash of formats in the following experiments: using N-terminal fusion of the degron tag for Erg20p and Acc1p and a C-terminal fusion for Hxk2p. If this was done for reasons other than ease of cloning, some explanation would be helpful. Moreover, the extra fast response elicited by inclusion of an NLS in the CUP1-AID tag appears not to be taken advantage of in any of the subsequent experiments.

It is also not entirely clear why and where strains with Tir1 only or the Tir1-Skp1 fusion was used.

A number of typos:

Line 205: FPP, geranyl should presumably be FPP, farnesyl

Line 247-256/fig 2c,d: Where is the minus NAA control?

Fig 2d: does LIM141 reach the same value as LIM401 eventually, ie. values at 120hrs?

Fig 2e: dark blue index square should be 96h

Line 264: typo: terpenoids

Line 266: typo: alcohols

Figure 2: Why was NAA inducer added at the seemingly rather early timepoint of 10 hrs. Since the AID system is fast and efficiently induced, spiking at a later stage (eg. 24 hrs) may give even better results. Or have different induction time-points already been tried and was 10 hrs the optimal? As mentioned in the conclusions, the AID strategy resulted in a lower limonene titre compared to a constitutive N-terminal degron approach, and this may well be due to 'spiking' too early.

Line 324: promoter-driven

Hxk2p section: The surprising findings of an early advantageous phenotype with the AID-CUP1-HXK2 strain even in the absence of NAA would appear to suggest that the tag modification by itself already affects folding or stability of Hxk2p. This highlights the need to investigate/profile protein levels directly by Western blot, as per earlier comment.

In the supplementary data a curious dichotomy between low and high producing NLD128 strains is mentioned, potentially due to genetic variation at other loci. This observation should briefly be mentioned in the main text, with reference to the supplementary data.

I find figure 3b very unclear and suggest that this should be redesigned, or at the least a proper explanation of the scheme should be given in the figure legend. There is also no figure legend for figure 3c. In figure 3d-l, where strain NLD128-1 is compared to strain NLD401 the data for NLD401 without NAA appears to be lacking?

The use of AID tag for decoupling of growth and metabolism is a neat concept and indeed seems to result in some improvement of nerolidol production. However, I have a few issues with figure 4: In panels c-j of this figure NAA is added at 24hrs, leading to growth arrest at 48hrs. In figure 4b however, incongruously, NAA is present from 0hrs, making it rather uninformative. It is also confusing that no growth whatsoever is observed here, while the suggestion from the other panels is that a delay period of continuing growth for 24hrs exists.

In figure 4c-j data is missing for the NLD138 and NLD401 strains without addition of NAA. This would be important data to demonstrate that the observed effects are due to NAA induced depletion of Acc1p.

Confusingly the authors have decided in this experiment to analyse effect of NAA addition at 24hrs (and at 0 hrs when growing on ethanol). The data however suggest that AID induction at 24 hrs, resulting in growth arrest around 48hrs, seems too late to observe a dramatic effect as this brings the growth curve already close to the stationary phase.

Addition at 0hrs (confusingly done with ethanol as sole carbon source) on the other hand is clearly too early as very little growth occurs.

I would suggest a repeat of this experiment with NAA addition at 10hrs. Alternatively, if good reasons exist for these choices, perhaps the authors can explain why they did not choose an intermediate timepoint for induction, eg. 10 hrs, as in the experiment of figure 2 (where contrastingly, in my opinion a timepoint of 24hrs would have been more suitable).

Reviewer #2:

Remarks to the Author:

Lu et al. report the development of an auxin based degron switch for dynamic control in *S. cerevisiae*, and demonstrate the utility of this methodology in several metabolic engineering applications. First of all I would like to congratulate the authors for the development of a tightly controlled degradation switch, with minimal changes to basal expression levels and effective rapid

degradation of target proteins. Overall the manuscript is well written and demonstrates a useful tool for dynamic control in *S. cerevisiae*.

Overall there are a few major points for the authors to address and a minor points.

Major Points.

1) The authors skip over the role of the process in these studies. There is an underlying dynamic shift in these cultures, that is independent of the switch, ie the diauxic shift accompanying glucose depletion, yet this is only stated as a mechanism to induce the gal promoters (as far as I can find). It would be very helpful to specifically call this out in figures, and place induction of the auxin switch in context of the overall culture dynamics. It would also be important to emphasize whether this method or the results are reliant on production post glucose depletion or whether the same results can be obtained during growth on a sugar substrate. This is important to understand the general applicability of the methodology or if it is constrained, or optimal only under a certain limited set of metabolic states and process conditions. This is highlighted by the results in Figure 4, where ethanol is used as a sole carbon source and the results elsewhere in the work demonstrating some "toxicity" in exponential phase. Is the timing of inducer addition relative to glucose depletion an important variable?, if so what is the optimal induction time and why, and why was the current induction time chosen.

2) While artificial inducers enable tight control over switches, they can also relegate many of these systems to "toy" studies demonstrating proof points, yet not translating to any potential commercial applications. As metabolic engineering is at heart aimed at an end goals to actually produce products, it would be good for the authors to be more specific about how they see the application/use of this switch. The authors seem to indicate this switch has a role in engineering applications as well as in studying metabolism, while the latter is clear the former is not. Perhaps the authors intend to use this switch for rapid prototyping and then engineer more commercially relevant strains? It would be good to discuss how this switch can best be used.

3) The discussion is primarily a recap o the results, and does not firmly put the work in context and outline how this work fits into the field and what future work is needed to build upon these results. For example, while the decoupling of growth from production led to an increase in product it was modest when compared to the control and the other relative changes observed when using the switch. There have been several papers reviewing and modeling the potential of decoupled metabolism. How do these results compare to these studies, is there a reason why the results observed were modest, what else should be pursued? Similarly the authors (as demonstrated by the work with hexokinase) highlight that dependent on the product and context the impact of this switch and dynamic alterations may be minimal compared to other overriding challenges. This is why they chose not to pursue limonene in the hexokinase portion of this work. Did the authors try this approach with limonene, if so what were the results, is nerolidol a unique example where the impact of the switch is highlighted? How would one in the field choose which products may benefit from this approach and which have bigger orthogonal challenges. How does this approach compare to other demonstrated switches in the field, what are the advantages or special use cases.

Minor points,

1. There are numerous minor grammatical issues with the manuscript. For example in the abstract the authors start with "First" to describe the first application, which should be followed by Second, ...Third, ... but it is not. This requires someone to manually try to break the abstract apart.

Responses to Reviewers' Comments

REVIEWER COMMENTS

Reviewer #1 (Remarks to the Author):

NCOMMS-20-31830

An auxin-mediated protein depletion switch for metabolic perturbation in yeast
Zeyu Lu, Bingyin Peng, Birgitta E. Ebert, Geoff Dumsday, Claudia E. Vickers.

Saccharomyces cerevisiae is an important platform strain for the production of various biofuels, biochemicals, and biopharmaceuticals. A large variety of industrially useful compounds are part of the isoprenoid family and use the same initial pathway to produce the requisite precursors. In recent times the synthetic biological approach of metabolic pathway engineering has become the method of choice for optimised production of selected products. Part of this approach is the re-direction of metabolic flux towards the desired pathway. Deletion of genes for enzymes that would direct precursors towards alternative metabolic routes is commonly used, but problematic when absence of the enzyme is detrimental to growth or survival of the yeast, while conditional deletion can be slow and ineffective depending on the protein.

Here the authors demonstrate that the use of the well-established Auxin inducible degron system can provide an effective alternative. The AID has become a favoured inducible protein degradation system due to its fast and efficient inducibility by the orthologous plant hormone auxin or its analogues, even though some constitutive leakage has been observed in several studies.

This is a well written manuscript describing several well-designed experiments, with a few reservations as mentioned below. Together the findings of this manuscript achieve to highlight the promising potential of the auxin inducible degron system for metabolic engineering and show it can be smart solution to diverse aspects such as flux redirection and the decoupling of growth and production.

Response: *Thank you for these comments*

Q1: Interestingly the authors have used a different title for the supplementary information file which in my opinion provides a better coverage of the contents of the manuscript.

Response: *We agree with that the more descriptive title from the Suppl. is better and have revised the title and. The new title we propose is: "Auxin-mediated protein depletion in metabolic engineering: flux redirection, metabolic modulation, and growth arrest in terpene-producing yeast".*

Q2: In the first part of this study the authors set up the AID system for their yeast strains and play around with a few construct formats to find the optimal set-up based on loss of fluorescence of AID tagged yEGFP fusion proteins. Interestingly fusion with CUP1 appears to prevent premature loss of fluorescence, while three other fusion partners make things worse. Whether this is due to increased basal protein level/stability or enhanced solubility remains unclear and whether CUP1 stabilises the AID-tag moiety, yEGFP protein or the combination of those. More importantly I wonder if the CUP1 effect is a general improvement of the AID-tag or, as would seem likely, that the effectiveness of the fusion is particular to each specific protein. Therefore, the question arises whether the CUP1-AID fusion is equally effective on the specific metabolic enzymes tagged in this study. It would be

informative to see comparative profile of the abundance of the various tagged and untagged proteins over time in the presence or absence of inducer by Western blot.

Response:

There are several questions embedded in this paragraph and we have answered them under the headings below.

Unfortunately, we did not add any immuno-tags in our designs to facilitate abundance measurements by western blot. Doing this experiment would therefore require purchase and optimisation of bespoke antibodies – an expensive and time-consuming process – or redesign of constructs by adding these tags, which might themselves interfere with the outcome of the experiment. Instead, we performed a limited study using proteomics, and have also drawn some conclusions about the efficacy of the approach based on observations from the current experimental dataset.

Firstly, we characterised physiological effects caused by auxin-mediated protein degradation (Figure 2-4). In all three cases (Erg20, Hxk2, and Acc1), adding auxin led to dramatic metabolic effects, for example, severe growth inhibition for depletion of Erg20 and Acc1, metabolic redirection in Erg20 case, lifting glucose repression in Hxk2 case, and growth arrest in Acc1 case. These examples demonstrate that the CUP1-AID tag is effective across a variety of different proteins. We also showed that tagging Cup1-AID to Hxk2 led to significant metabolic modulation even without addition auxin to trigger Hxk2p depletion (Figure 3).

To directly address the question of ‘whether CUP1-AID is equally effective on reduction of auxin-independent degradation of the specific metabolic enzymes’, we examined performance of Erg20 and Acc1 strains tagged with Cup1-AID without auxin addition. Surprisingly, we observed significant metabolic redirection towards monoterpene (limonene) in the Cup1-AID-Erg20 strain even in the absence of auxin. There was a slight growth difference and small (although insignificant) increase in nerolidol production for the Cup1-AID-Acc1 strain. Both of these examples therefore indicate indirectly that protein degradation happens upon AID tagging even in the absence of auxin.

To examine this further, we did an extra experiment: comparative proteomics by LC-MS/MS to analyse Hxk2 protein abundance in the Hxk2-AID-Cup1 strain compared to Hxk2 WT strain. This showed a 38% decrease of Hxk2 abundance compared to wildtype strain ($p = 0.26$) (new data in Supplementary Figure S5). This confirms that tagging with the AID tag results in a decrease in the tagged protein - consistent with the results we observed when yEGFP was used as the reporter.

Combining the physiological observations, quantitative data, and proteomics data, it is clear that the tag is effective across a range of target proteins to deliver the required engineering outcomes. It is likely that the quantum of effect is variable across target proteins and under differing conditions due to idiosyncrasies of the test systems. Therefore, it is advisable to users to test each protein. We have added a statement to this effect in the Conclusions, and we have clarified these considerations throughout the main text as follows (new text in the manuscript is highlighted in yellow below):

- (a) We added the results of physiological performance of Cup1-AID-Erg20 strain and Cup1-AID-Acc1 strain under the conditions without auxin addition in main text*
- (b) We added the following text in the Discussion: ‘We postulate that these broad-scale metabolic responses resulting from basal decrease of protein abundance of Erg20p and Hxk2p and/or negative effects on normal protein function upon modification with AID tag,*

like that modification on Hxk2p may influence its function in glucose sensing networks⁶³. These results suggest that modifying a target protein by tagging may lead to a dramatic regulatory effect in some cases. ‘

- (c) We added an additional paragraph in discussion to emphasise that ‘It is necessary to emphasize that, in the three examples (Erg20p, Hxk2p, and Acc1p) we tested, tagging target protein with the AID tag had idiosyncratic effects for individual proteins, making prediction of the phenotype challenging. Despite this, tagging did not dramatically influence growth until auxin NAA was added to trigger the depletion of target protein.’
- (d) We added the following sentence in the discussion: ‘It is likely that the quantum of effect is variable across target proteins and under differing conditions due to idiosyncrasies of the test systems. Therefore, it is advisable to users to test each protein.’
- (e) In addition, we thought it might be helpful to readers to clarify further why we tested fusion partners to reduce the basal degradation in the first place. We revised the following paragraph by adding the reasons in the main text in result section 1:

‘To address potential causes of decreased GFP fluorescence in the AID*-tagged yEGFP strain, we investigated the effect of adding tags to the N-terminus of the AID* peptide. This approach might contribute to minimising auxin-independent degradation by insulation AID* from the terminus, which was observed previously when 9 myc epitopes was used²⁵. However, repeated myc sequences could be difficult for genetic manipulation. We therefore tested other fusion partners: thioredoxin (TRX1), superoxide dismutase (SOD1), monomeric red fluorescent protein (mRFP), and metallothionein (CUP1). ‘

=====

In the next sections the authors apply the AID system to three different enzymes representing three aspects of metabolic engineering, building on their knowledge and in-house selection of engineered yeast strains and constructs, and appear to achieve improvements in each case. Below are a few, mostly minor comments on these sections:

Q3: Firstly, despite their efforts to establish the optimal format for their AID set-up in Fig1, they appear to use a mish-mash of formats in the following experiments: using N-terminal fusion of the degron tag for Erg20p and Acc1p and a C-terminal fusion for Hxk2p. If this was done for reasons other than ease of cloning, some explanation would be helpful. Moreover, the extra fast response elicited by inclusion of an NLS in the CUP1-AID tag appears not to be taken advantage of in any of the subsequent experiments. It is also not entirely clear why and where strains with Tir1 only or the Tir1-Skp1 fusion was used.

Response:

There are several questions in this paragraph that we have answered as follows (text highlighted in yellow is new text in the manuscript):

1. Question: was the use of an N-terminal fusion of the degron tag for Erg20p and Acc1p and a C-terminal fusion for Hxk2p due to ease of cloning or another reason?

Response: There were several reasons that the tagging approach used was variable; these related to avoiding interference with protein function. We added the following information to explain the reasons for N-terminal fusion of degron tag for Erg20p and Acc1p and a C-terminal fusion of Hxk2p:

'Because the C-terminal tail of FPP synthase is important for the binding of substrate IPP in its catalytic pocket⁴⁰, Erg20p was tagged with N-terminal Cup1-AID through CRISPR/Cas9-mediated genome editing (strain o141; Table S2).'*

'For auxin-mediated depletion of Hxk2p, we fused the AID-Cup1p tag to the Hxk2p C-terminus through CRISPR/Cas9-mediated genome editing (strain o138; Table S2). The C-terminus was chosen because the N-terminal residues of Hxk2p are important for its role in glucose signalling⁵³⁻⁵⁵ and it was critical not to interfere with this function. The C-terminal of Hxk2p is important for catalytic activity⁵⁵, which for the purposes of this experiment is less important and can be complimented by the activity of other isoforms.'*

'For auxin-mediated depletion of Acc1p, we fused the Cup1p-AID tag to the Hxk2p N-terminus through CRISPR/Cas9-mediated genome editing (strain o128; Table S2). We targeted the N-terminal because the N-terminal residues of Acc1p are exposed outside and the C-terminal residues are in a pocket⁶².'*

2. Question: the extra fast response elicited by inclusion of an NLS in the CUP1-AID tag appears not to be taken advantage of in any of the subsequent experiments.

Response: NLS is a nuclear localisation signal. NLS addition is not an advantage for this system because we are working with proteins which are naturally in cytosol and/or nucleus. To make it more clear why NLS is added to test yEGFP-AID-CUP1-NLS degradation, we revised the beginning of that paragraph as the following:

'Many yeast proteins are localised in nucleus or simultaneously in nucleus and cytosol. In order to test effectiveness of the engineered system for nuclear-localised proteins, we investigated localisation of the construct using a nuclear localisation sequence (NLS).'

'These showed that auxin-inducible protein degradation machineries functioned for degradation of target protein localised in both cytosol and nucleus in yeast.'

3. Question: It is also not entirely clear why and where strains with Tir1 only or the Tir1-Skp1 fusion was used.

Response: We added the following information to explain the reasons why Tir1-Skp1 was used:

'For minimal potential effects on metabolism, caused by auxin-independent degradation of target enzymes, we then used the compromised SKP1-TIR1 system, with Cup1p-AID or AID*-Cup1p as the tag, rather than TIR1 system that led to faster depletion of target protein, in the following studies.'*

Q4: A number of typos:

Line 205: FPP, geranyl should presumably be FPP, farnesyl

Response: *We have revised it. Thanks.*

Line 247-256/fig 2c,d: Where is the minus NAA control?

Response: *We added minus NAA control for CUP1-AID-ERG20 strain.*

Fig 2d: does LIM141 reach the same value as LIM401 eventually, ie. values at 120hrs?

Response: *We added two control experiments for LIM141 w/o NAA addition and LIM141 with NAA added at 24 hour. Y-FAST induction could reach to the value in LIM401.*

Fig 2e: dark blue index square should be 96h

Response: *Figure 2e has been updated.*

Line 264: typo: terpenoids

Response: *We have revised it. Thanks.*

Line 266: typo: alcohols

Response: *We have revised it. Thanks.*

Q5: Figure 2: Why was NAA inducer added at the seemingly rather early timepoint of 10 hrs. Since the AID system is fast and efficiently induced, spiking at a later stage (eg. 24 hrs) may give even better results. Or have different induction time-points already been tried and was 10 hrs the optimal? As mentioned in the conclusions, the AID strategy resulted in a lower limonene titre compared to a constitutive N-terminal degnon approach, and this may well be due to 'spiking' too early.

(1) There were several reasons that we chose to add the NAA inducer at 10 hour. To make this clear in the manuscript, we revised the paragraph as the following:

'LIM401 and LIM141 were characterised in dodecane-overlayered MES-buffered YNB-glucose medium. We first tested the performance of both strains with NAA added at 10 hours to 1 mM final concentration. We presumed: (1) that either residual Erg20p activity over the remaining exponential phase, or residual sterol pools in the cell generated during growth prior to addition of NAA, was sufficient to sustain growth prior to entering stationary phase; (2) that the residual FPP synthase activity of the ERG20^{N127W} mutant may sustain sufficient pathway flux to supplement degradation of native ERG20^{3p}; and (3) that triggering depletion of Erg20p in an early phase may prevent the accumulation of FPP after the induction of GAL promoters-controlled MVA pathway upon diauxic shift at 13-to-15 hour.'

- (2) We also performed an additional experiment to address the two questions regarding the timing of the NAA addition, i.e. (1) 'spiking at a later stage (eg. 24 hrs) may give even better results' and (2) 'As mentioned in the conclusions, the AID strategy resulted in a lower limonene titre compared to a constitutive N-terminal degron approach, and this may well be due to 'spiking' too early'. We performed the experiment of adding NAA at 24 hour and added the new results into results section, the last two paragraphs in section 'Flux redirection: degradation of farnesyl pyrophosphate improves monoterpene (limonene) production'.

In a short summary, spiking at a later stage (at 24 hour) is too late to prevent the accumulation of Erg20 downstream product and to redirect more metabolic flux toward limonene production. Consequently, the strain did not perform as well under this condition.

- (3) Finally, we also included the following in the results section to contextualise these findings:

For a better understanding of the auxin-mediated protein degradation tool in metabolic engineering, we further tested the LIM141 strains without NAA addition or with NAA added at 24 hour to trigger Erg20p depletion. Adding NAA at 24 hour did not dramatically change the terpene production profiles in LIM141, though it did lead to a slight (but not statistically significant) decrease in the induction level of Y-FAST (Figure 2d), compared to the conditions without NAA addition. Unexpectedly, in the absence of NAA addition to trigger Erg20p depletion, LIM141 showed increased production of monoterpenes limonene and geraniol and decreased production of farnesol, compared to ERG20-reference strain LIM401 (Figure 2e-g). This suggests that there auxin-independent flux redirection occurs in strain LIM141. It further indicates that there are unpredictable and non-specific perturbation effects which can result from the introduction of the auxin-inducible protein degradation system. Such perturbation effects might be caused by the decreased efficiency of Erg20p expression or catalytic function (wither of which might be a result of the degradation tag, which might influence transcription efficiency, translation efficiency, or catalytic function); or could also be due to non-specific auxin-independent degradation of Erg20p. The latter cannot be excluded because the basal degradation had not been fully eliminated in the current system (as evidenced by the yEGFP reporter experiments – see Figure 1).

Strain LIM141 exhibited similar physiological features both in the absence of NAA and when NAA was added at 24 hour, compared to when NAA was added at 10 hour. The features included: (1) improved growth during the post-exponential phase (Figure 2c); (2) production of limonene primarily in the first 48 hours rather than in the period from 48 hour to 96 hour, with limonene titres being higher at 48 hour than the 10 hr experiment (Figure 2e); and (3) prenyl alcohols were produced consistently during the first 48-hour period and during the period from 48 hour to 96 hour. Overall, we observed lower geraniol titres, higher farnesol titres, and higher geranylgeraniol titres (Figure 2f-h). This indicates that it is important to constrain the flux towards FPP production during exponential growth (before the diauxic shift) to deliver improved monoterpene production. It also suggests the possibility that prenyl pyrophosphate products downstream of ERG20 (i.e., FPP or GGPP) can inhibit upstream pathways and/or limonene synthase through an as-yet unknown feedback

mechanism (presumably through competitive inhibition; but this is yet to be investigated).

Q6: Line 324: promoter-driven

Response: *We have revised it. Thanks.*

Q7: Hxk2p section: The surprising findings of an early advantageous phenotype with the AID-CUP1-HXK2 strain even in the absence of NAA would appear to suggest that the tag modification by itself already affects folding or stability of Hxk2p. This highlights the need to investigate/profile protein levels directly by Western blot, as per earlier comment.

Response: *See the responses to Q2 above.*

Q8: In the supplementary data a curious dichotomy between low and high producing NLD128 strains is mentioned, potentially due to genetic variation at other loci. This observation should briefly be mentioned in the main text, with reference to the supplementary data.

Response: *The information has been added into the main text, as following:*

'To evaluate the effect of Hxk2p depletion on nerolidol production, a HXK2-AID-CUP1 nerolidol-producing strain NLD128 was developed (Supplementary Result 2 & Supplementary Figure S3). However, there were two types of NLD128 clones: one type showing higher Y-FAST expression and higher nerolidol production than another type (Supplementary Result 2 & Supplementary Figure S3). HXK2 modifications in both types of clones was confirmed by DNA sequencing, and the sequence was as being designed. The causes of this variation might be due to the variations in plasmid copy number or at other genetic loci. In the following study, we chose a high-nerolidol-production clone NLD128-1 to further examine the effects of Hxk2p depletion.'*

Q9: I find figure 3b very unclear and suggest that this should be redesigned, or at the least a proper explanation of the scheme should be given in the figure legend. There is also no figure legend for figure 3c. In figure 3d-l, where strain NLD128-1 is compared to strain NLD401 the data for NLD401 without NAA appears to be lacking?

Response:

(1) *Figure 3b has been re-coloured with addition of a note. The legend for Figure 3 was revised to: 'b. Hxk2p in Snf1p-dependent glucose repression on GAL promoters via Mig1p and respiration. #, In HXK2-AID*-CUP1 strain NLD128-1, auxin mediates Hxk2p depletion, and in HXK2-wildtype strain NLD401 Hxk2p is not regulated by auxin.' A legend for Figure c was included: 'c-l, characterisation of strain NLD401 and strain NLD128 with or without addition of 1 mM NAA at 0 hour and in precultures'.*

(2) *We did not test carbohydrate metabolism for NLD401 w/o NAA addition, whereas we referred to our previous published data in the similar strains and wildtype strains and the current data in NLD401 with NAA addition to perform a preliminary comparison of carbohydrate metabolism, as below:*

'This was accompanied by faster carbon uptake, both on glucose and ethanol, (Figure 3e and 3h) compared to NLD401 with NAA added. Carbon uptake in this strain was also faster than in previous nerolidol producing strains and in wild type S. cerevisiae CEN.PK^{27,33}.'

- (3) Growth, Y-FAST, and nerolidol data for NLD401 w/o NAA was in Supplementary Figure S2, which was missing due to the mistake of uploading a different version of supplementary document, although it was included in the main text as following:

'Induction of the GAL promoter controlling Y-FAST-2A-AcNES1 (as inferred from the Y-FAST fluorescence) and nerolidol production in the reference HXK2 strain NLD401 was no different in the presence or absence of NAA (Figure 3f and 3g & Supplementary Figure S2).'

To address the mistake, we have updated the Supplementary document, and added 'The growth, Y-FAST fluorescence, and nerolidol production in strain NLD401 without NAA addition is available in Supplementary Figure S2' in Figure 3 legend.

Q10: The use of AID tag for decoupling of growth and metabolism is a neat concept and indeed seems to result in some improvement of nerolidol production. However, I have a few issues with figure 4:

In panels c-j of this figure NAA is added at 24hrs, leading to growth arrest at 48hrs. In figure 4b however, incongruously, NAA is present from 0hrs, making it rather uninformative. It is also confusing that no growth whatsoever is observed here, while the suggestion from the other panels is that a delay period of continuing growth for 24hrs exists.

In figure 4c-j data is missing for the NLD138 and NLD401 strains without addition of NAA. This would be important data to demonstrate that the observed effects are due to NAA induced depletion of Acc1p.

Response:

- (1) Growth, Y-FAST, and nerolidol data for NLD401 w/o NAA was in Supplementary Figure S2, which was missing due to the mistake of uploading a different version of supplementary document. To address the mistake, we have updated the Supplementary document, and added 'The growth, Y-FAST fluorescence, and nerolidol production in strain NLD401 without NAA addition is available in Supplementary Figure S2' in Figure 4 legend.
- (2) The reviewer comments that 'In figure 4b however, incongruously, NAA is present from 0hrs, making it rather uninformative' – however, there is no experiment with NAA added at 0 hour in figure 4b. We are unsure what the reviewer means by this comment.
- (3) A reference experiment for NLD138 without addition of NAA was added into figure 4c-f, which was sufficient to demonstrate that the observed effects are due to NAA induced depletion of Acc1p. The main text was updated accordingly by adding the following information:
'Under the conditions without NAA addition, strain NLD138 (Figure 4c-e) showed a decreased growth, slightly faster Y-FAST induction during the post-exponential growth phase, and produced similar amount of nerolidol at 72 hour, compared to strain NLD401

(Supplementary Figure S2c, S2f-g). The slight changes in growth and Y-FAST induction showed an auxin-independent perturbative effects in the CUP1-AID-ACC1 strain.'*

Q11: Confusingly the authors have decided in this experiment to analyse effect of NAA addition at 24hrs (and at 0 hrs when growing on ethanol). The data however suggest that AID induction at 24 hrs, resulting in growth arrest around 48hrs, seems too late to observe a dramatic effect as this brings the growth curve already close to the stationary phase.

Addition at 0hrs (confusingly done with ethanol as sole carbon source) on the other hand is clearly too early as very little growth occurs.

I would suggest a repeat of this experiment with NAA addition at 10hrs. Alternatively, if good reasons exist for these choices, perhaps the authors can explain why they did not choose an intermediate timepoint for induction, eg. 10 hrs, as in the experiment of figure 2 (where contrastingly, in my opinion a timepoint of 24hrs would have been more suitable).

Response:

- (1) We chose to add NAA at 24 hour in order to test the growth-arrest effects after diauxic shift. To respond to the reviewer's question about what would happen in the Cup1-AID-Acc1 strain NLD138 under the conditions without NAA addition or with NAA added at 10 hour, we performed additional strain characterisation. The results have been added in results section in 'Growth Arrest: depletion of Acc1p arrests growth but not production'.*

To summarise: triggering Acc1 depletion at 10 hour led to growth arrest after 24 hour and the stalling of GAL promoter inductions. Consequently, the cells did not reach to the high-production state shown by the cells with growth arrest triggered by NAA addition at 24 hour.

- (2) Characterisation on ethanol was performed with addition of NAA at 0 hour because we were seeking to understand the effects of growth arrest when the GAL promoters were fully de-repressed. To make this clearer in the manuscript, we revised the first sentence of the mentioned paragraph like so:*

'For a better understanding of the effects of growth arrest in NLD138 under the conditions with fully de-repression of GAL promoters, we further characterised strain NLD138 and the reference strain NLD401 with ethanol as the sole carbon source, and with NAA added at 0 hour.'

- (3) By combination of the results with NAA added at 10 hour or 24 hour in cultivation on glucose and the results with NAA added at 0 hour in cultivation on ethanol, we then reached to the following conclusion (which we have added to the manuscript):*

In flask cultivation with 20 g L⁻¹ glucose as the carbon source, no difference was observed between the reference strain NLD401 (wild type ACC1) and strain NLD138 during the exponential growth phase (Figure 4c & Supplementary Figure S2c). In the absence of NAA, strain NLD138 showed a decreased growth, slightly faster Y-FAST induction during the post-exponential growth phase, and produced similar amount of nerolidol at 72 hour, compared to strain NLD401 (Figure 4c-e, Supplementary Figure S2c, S2f-g). These changes demonstrate an auxin-independent perturbation in the CUP1-AID-ACC1 strain.*

When NAA was added at 10 hour to trigger Acc1p depletion, strain NLD138 showed growth arrest after 24 hour (Figure 4c). Y-FAST fluorescence plateaued after 48 hours, suggesting that induction stopped at 48 hours (Figure 4d). After growth arrest, cells were still metabolically active. The nerolidol titre at 72 hours was $\sim 0.76 \text{ g L}^{-1}$, 43% of the titre in the absence of NAA addition. Consistent with this, the specific nerolidol production rate was induced to lower levels compared to the reference strain NLD401 between 24 hour and 72 hour (Figure 4f). This might result from stalling GAL promoter induction (the pathway genes for nerolidol synthesis are controlled by GAL promoters).

NAA addition at 24 hour led to growth arrest in strain NLD138 after 48 hour, while strain NLD401 continued to grow (similar to the growth profile in the absence of NAA; Figure 4c & Supplementary Figure S2c). Depleting Acc1p after 24 hour did not significantly influence Y-FAST induction (Figure 4d), indicating that GAL promoter-driven expression was relatively unperturbed; however, nerolidol production was significantly improved, with the titre increased by $\sim 36\%$ to $\sim 2.2 \text{ g L}^{-1}$ (Figure 4e). The specific nerolidol production rate was the same during the first 24 hours, but increased by $\sim 50\%$ during the period from 24 hour to 72 hour, compared to strain NLD401 (Figure 4f). This indicates that the growth-arrested cells maintained a superior metabolic state for nerolidol production when Acc1p depletion was triggered after the diauxic shift (with the induction of GAL promoters unperturbed).

For a better understanding of the effects of growth arrest in NLD138 the GAL promoters were fully de-repressed, we further characterised strain NLD138 and the reference strain NLD401 with ethanol as the sole carbon source, and with NAA added at 0 hour. Neither of the strains show exponential growth (Figure 4g), possibly as a result of metabolic imbalance due to the strong induction of GAL promoter-controlled mevalonate pathway (as shown by the induction of Y-FAST fluorescence; Figure 4h). However, the growth of strain NLD138 was much more severely impaired. Induction of Y-FAST halted after 24 hours in strain NLD138, whereas it continued in strain NLD401. Although growth was severely impaired, strain NLD138 cells were metabolically active, as shown by nerolidol production (Figure 4i). The specific nerolidol production rates in strain NLD138 were similar to the rates in strain NLD401. These results indicate that when cells are grown on ethanol with nerolidol synthetic pathways (including the mevalonate pathway) induced to a high level, Acc1p depletion did not affect acetyl-CoA flux redirection. The specific nerolidol production rates in growth-arrested NLD138 cells did not decline dramatically after the more prolonged incubation (Figure 4j), and showed survival rates similar to the reference without growth arrest (Supplementary Table S6).

These results demonstrate that growth-arrested cells from auxin-inducible depletion of Acc1p are metabolically active, and that nerolidol productivities in the growth-arrested cells correlate with the induction state of nerolidol synthetic pathways.

Reviewer #2 (Remarks to the Author):

Lu et al. report the development of an auxin based degron switch for dynamic control in *S. cerevisiae*, and demonstrate the utility of this methodology in several metabolic engineering applications. First of all I would like to congratulate the authors for the development of a tightly controlled degradation switch, with minimal changes to basal expression levels and effective rapid degradation of target proteins. Overall the manuscript is well written and demonstrates a useful tool for dynamic control in *S. cerevisiae*.

Response: *Thank you for these encouraging comments!*

Overall there are a few major points for the authors to address and a minor points.

Major Points.

Q12) The authors skip over the role of the process in these studies. There is an underlying dynamic shift in these cultures, that is independent of the switch, i.e. the diauxic shift accompanying glucose depletion, yet this is only stated as a mechanism to induce the gal promoters (as far as I can find). It would be very helpful to specifically call this out in figures, and place induction of the auxin switch in context of the overall culture dynamics. It would also be important to emphasize whether this method or the results are reliant on production post glucose depletion or whether the same results can be obtained during growth on a sugar substrate. This is important to understand the general applicability of the methodology or if it is constrained, or optimal only under a certain limited set of metabolic states and process conditions. This is highlighted by the results in Figure 4, where ethanol is used as a sole carbon source and the results elsewhere in the work demonstrating some "toxicity" in exponential phase. Is the timing of inducer addition relative to glucose depletion an important variable?, if so what is the optimal induction time and why, and why was the current induction time chosen.

Response:

- (1) We have made the revisions in Figure 2a, 3a, and 4a to indicate GAL promoters being 'glucose-depletion-inducible'. In legend of Figure 3 and 4, we added a sentence to indicate Y-FAST was under the control of the GAL2 promoter.*
- (2) Reviewer 1 also raised the similar questions, regarding to the variation of induction time of auxin-mediated depletion of target proteins. Please see our responses to Q5 and Q11 above.*

Q13) While artificial inducers enable tight control over switches, they can also relegate many of these systems to "toy" studies demonstrating proof points, yet not translating to any potential commercial applications. As metabolic engineering is at heart aimed at an end goals to actually produce products, it would be good for the authors to be more specific about how they see the application/use of this switch. The authors seem to indicate this switch has a role in engineering applications as well as in studying metabolism, while the latter is clear the former is not. Perhaps the authors intend to use this switch for rapid prototyping and then engineer more commercially relevant strains? It would be good to discuss how this switch can best be used.

Response:

Thank you for this comment. A discussion on how useful this tool is in an industrial setting, contextualising to the data presented, is very appropriate. We agree that the utility of this tool in an

industrial setting would require further examination, and that at this stage the tool is most useful for rapid prototyping. To address Reviewer's concerns about the way to use the current tool, we added the following text to the Discussion and Conclusion:

Discussion:

'It is necessary to emphasize that, in the three examples (Erg20p, Hxk2p, and Acc1p) we tested, tagging target protein with the AID tag had idiosyncratic effects for individual proteins, making prediction of the phenotype challenging. Despite this, tagging did not dramatically influence growth until NAA was added to trigger the depletion of target protein. Depletion of these three proteins not only lead to improved production of the target terpene products, but also caused broad phenotypic responses in yeast cells, including responses in metabolism, gene expression, and cell growth (Figure 2-4). We also found that timing of induction for protein depletion is a critical consideration for the process. For example, triggering Erg20p depletion in the exponential phase (at 10 hour) resulted in significant flux redirection toward limonene production (Figure 2). However, triggering Erg20p depletion in the post-exponential phase (at 24 hour) was less successful, presumably because FPP accumulation prior to depletion of Erg20p inhibited the upstream pathways. Triggering Acc1p depletion at 10 hour and at 24 hour resulted in growth-arrested cells with varying nerolidol productivities, which positively correlated with induction levels of GAL promoters (Figure 4). These results demonstrate that auxin-inducible protein degradation can be applied for a swift perturbation, and that removing the target protein at different stages delivers an improved understanding of the regulation of metabolism and other intracellular processes. This could be performed in combination with model-driven system biology studies⁶⁴⁻⁶⁶ to further examine metabolism.'

Conclusion:

'By developing strategies for flux redirection, metabolic modulation, and growth arrest, we showed that auxin-mediated protein degradation can be broadly applied in metabolic engineering for investigation of metabolic optimisation strategies. The approach provides a rapid, flexible loss-of-function perturbation switch. The method can also be used in systems biology studies for a better understanding of metabolism and metabolic regulation.

Application of this tool in an industrial setting will require further examination to determine feasibility. Here, we have demonstrated its use for rapid phenotyping of metabolic perturbations, which might be useful in a metabolic engineering context. Different products present different production challenges, and the ultimate choice of engineering approaches is dependent on the metabolic context of the desired product. The utility of the tools we have developed will need to be tested individually for each metabolic engineering project. They will have the most utility in cases where inducible down-regulation at the protein level is desirable - e.g., for very stable proteins, for proteins in essential pathways, and for decoupling production from metabolism. Further optimisation of auxin-inducible protein degradation tools might be achieved using other systems with a lower basal degradation and rapid depletion of target protein upon auxin addition. Similar systems have been reported recently in mammalian cells^{20,26}. In addition, integrating with auxin-mediated quorum-sensing mechanisms^{22,70} may trigger automatic regulation in metabolic engineering. Implementing Y-FAST as a tool, we observed single-cell level changes in the induction of the GAL promoter in terpene-producing strains with auxin-inducible degradation of an enzyme, which can instruct further studies to understand metabolic and genetic regulation in yeast.'

Q14) The discussion is primarily a recap of the results, and does not firmly put the work in context and outline how this work fits into the field and what future work is needed to build upon these results. For example, while the decoupling of growth from production led to an increase in product it was modest when compared to the control and the other relative changes observed when using the switch. There have been several papers reviewing and modeling the potential of decoupled metabolism. How do these results compare to these studies, is there a reason why the results observed were modest, what else should be pursued? Similarly the authors (as demonstrated by the work with hexokinase) highlight that dependent on the product and context the impact of this switch and dynamic alterations may be minimal compared to other overriding challenges. This is why they chose not to pursue limonene in the hexokinase portion of this work. Did the authors try this approach with limonene, if so what were the results, is nerolidol a unique example where the impact of the switch is highlighted? How would one in the field choose which products may benefit from this approach and which have bigger orthogonal challenges. How does this approach compare to other demonstrated switches in the field, **what are the advantages or special use cases.**

Response: *Thanks for highlighting that the discussion needs more work. Decoupling growth from production didn't lead to an increase in nerolidol production for the Acc1p experiments; however it did demonstrate that nerolidol production continues (and is not diminished) when growth is arrested. This shows that the cells are still metabolically active, which was the aim of the experiment in this case. We have clarified this by introducing text as described below. In addition, we have discussed our work in the context of other growth-arrest studies.*

The reviewer also notes that different products present different production challenges, and that the ultimate choice of engineering approaches is dependent on the metabolic context of the desired product. This is indeed quite true, and was indeed a driving factor in the selection of the example experiments we used to demonstrate our tool. We didn't try this approach with limonene, and are not able to do this experiment currently due to availability of capability. We agree that each product is likely to present idiosyncratic challenges, and thus will need to be tested individually as individuals pursue a metabolic engineering project. We have parameterised it here to a sufficient extent that other users will be able to determine whether or not it will have potential utility in their system. The tool we have developed is a powerful addition to the metabolic engineering and synthetic biology toolbox and while we would like to examine it in more contexts, ultimately, it will require the end user to examine it in their bespoke system to determine its use for that system. It will have the most utility in cases where inducible down-regulation at the protein level is desirable, that is, for very stable proteins or for proteins in essential pathways, and for decoupling production from metabolism.

In response to the reviewers' comments and to embed the considerations we have noted above, we have re-drafted as described below.

- (1) We improved the discussion section for the decoupling of growth from production, as following:

'Acc1p depletion caused growth arrest, but nerolidol productivity was not diminished (Figure 4). This demonstrates successful decoupling of growth and nerolidol production in this strain. Flux redirection by Acc1p depletion was only observed under specific conditions when heterologous nerolidol synthesis was induced to a medium level in ethanol-growth phase in flask cultivation on glucose (Figure 4f) rather than when nerolidol synthesis was induced to the maximum level in flask cultivation on ethanol (Figure 4j). Restricting biomass production by limiting

flux through an essential pathway has previously been successful in redirecting carbon flux towards production of interested heterologous metabolites in different organisms. Examples include improving production of myo-inositol derivatives by restricting glycolysis⁶⁷, improving shikimate production by restricting the downstream shikimate pathway essential for aromatic amino acid synthesis⁶⁷, improving fatty acid production by restricting leucine synthesis or nitrogen starvation⁶⁸, and improving carbohydrate production by restricting the tricarboxylic acid cycle⁶⁹. In our case, we presumed that a decrease in Acc1p activity increased acetyl-CoA availability for redirection into the mevalonate pathway, leading to improved nerolidol titre and improved specific production rate in flask cultivation on glucose. The observed effect of improved nerolidol production was modest and required specific conditions.

- (2) We did not try Hxk2p-depletion or Acc1p-depletion approach in limonene production, because the primary bottleneck for limonene production is the inefficient limonene synthase enzyme. We have addressed this problem in the Result section 3, as below:

‘Monoterpene production in yeast is inefficient due to poor catalytic activity of monoterpene synthases, resulting in conversion of excess GPP to geraniol in limonene producing strains (**Figure 2f**)³⁴. Accumulation of prenyl pyrophosphates is suspected to exert a toxic effect on cells⁵⁶. These adverse effects might impede evaluation of the full potential of metabolic engineering strategies. We therefore used yeast strains engineered for the synthesis of the sesquiterpene nerolidol, for which we have achieved substantial product titres^{33,34}, indicating a more efficient flux through to the target product.’

- (3) Discussion for the advantage or the potential way to use of the auxin switch was included in our response to Q13 above.

- (4) In the Conclusion, we added the following statement:

Different products present different production challenges, and the ultimate choice of engineering approaches is dependent on the metabolic context of the desired product. Thus, the utility of the tools we have developed will need to be tested individually for each metabolic engineering project. They will have the most utility in cases where inducible down-regulation at the protein level is desirable, that is, for very stable proteins or for proteins in essential pathways, and for decoupling production from metabolism.

Q15 Minor points,

1. There are numerous minor grammatical issues with the manuscript. For example in the abstract the authors start with “First” to describe the first application, which should be followed by Second, ...Third, ... but it is not. This requires someone to manually try to break the abstract apart.

Response:

We have revised the abstract accordingly. Thanks.

Reviewers' Comments:

Reviewer #1:

Remarks to the Author:

NCOMMS-20-31830

Auxin-mediated protein depletion in metabolic engineering: flux redirection, metabolic modulation and growth arrest in terpene-producing yeast.

The current manuscript presented here is a resubmission of the earlier manuscript: 'an auxin mediated protein depletion switch for metabolic perturbation in yeast'. In this revised version of the manuscript the authors have made several changes in response to reviewers' comments and they provide a very detailed and thorough rebuttal letter.

I believe the authors have made a good choice in deciding to change the title of the manuscript to the current one which provides a clear and appropriate description of the paper.

While the AID degron is a great system for the rapid degradation of target proteins, as indeed shown in this study, there are protein-specific idiosyncrasies associated with it as also observed here. The added paragraphs in the discussion highlighting these idiosyncrasies for each tagged protein, as well as the variable effects of the AID tag in the absence of auxin now provide a better/fairer reflection of the fact that there are potential target protein-specific drawbacks with its use and that modulations to the system (eg. the CUP1 'buffer') are also protein-specific. The inclusion of a few extra sentences in the text explaining the authors reasons for certain approaches/choices (e.g. lines 131-133; 178-181; 231) are also helpful to the reader.

The authors have carried out several extra measurements in response to my queries about the timing of NAA addition in some of the experiments. While the results from these experiments overall have not altered the original findings, I believe these additional tests/controls, as well as the provided explanations for the timing choices, nevertheless make good improvements to the soundness of the manuscript.

The addition of a short discussion on the subject of the use of the AID system as a prototyping tool versus its application in an industrial setting is also appreciated.